# A Distributional Perspective on Actor-Critic Framework

## Abstract

Recent distributional reinforcement learning methods, despite their successes, still contain fundamental problems that can lead to inaccurate representations of value distributions, such as distributional instability, action type restriction, and conflation between samples and statistics. In this paper, we present a novel distributional actor-critic framework, GMAC, to address such problems. Adopting a stochastic policy removes the first two problems, and the conflation in the approximation is alleviated by minimizing the Cramér distance between the value distribution and its Bellman target distribution. In addition, GMAC improves data efficiency by generating the Bellman target distribution through the Sample-Replacement algorithm, denoted by SR($\lambda$), which provides a distributional generalization of multi-step policy evaluation algorithms. We empirically show that our method captures the multimodality of value distributions and improves the performance of a conventional actor-critic method with low computational cost in both discrete and continuous action spaces, using Arcade Learning Environment (ALE) and PyBullet environment.

## 1 Introduction

The ability to learn complex representations via neural networks has enjoyed success in various applications of reinforcement learning (RL), such as pixel-based video gameplays (Mnih et al., 2015), the game of Go (Silver et al., 2016), robotics (Levine et al., 2016), and high dimensional controls like humanoid robots (Lillicrap et al., 2016; Schulman et al., 2015). Starting from the seminal work of Deep Q-Network (DQN) (Mnih et al., 2015), the advance in value prediction network, in particular, has been one of the main driving forces for the breakthrough.

Among the milestones of the advances in value function approximation, distributional reinforcement learning (DRL) further develops the scalar value function to a distributional representation. The distributional perspective offers various benefits by providing more information on the characteristics and the behavior of the value. One such benefit is the preservation of multimodality in value distributions, which leads to more stable learning of the value function (Bellemare et al., 2017a).

Despite the development, several issues remain, hindering DRL from becoming a robust framework. First, a theoretical instability exists in the control setting of value-based DRL methods (Bellemare et al., 2017a). Second, previous DRL algorithms are limited to a single type of action space, either discrete (Bellemare et al., 2017a; Dabney et al., 2018b;a) or continuous (Barth-Maron et al., 2018; Singh et al., 2020). Third, a common choice of loss function is Huber quantile regression loss which is vulnerable to conflation between samples and statistics without an imputation strategy (Rowland et al., 2019).

The instability issue is not present if a trainable policy is introduced, i.e., the evaluation setting of the Bellman operator is used, as shown by the convergence of distributional Bellman operator (Bellemare et al., 2017a). In addition, the general form of the stochastic policy gradient method does not assume a specific type of action space, e.g. discrete or continuous (Williams, 1988; 1992; Sutton et al., 1999). Because Wasserstein distance has biased sample gradients (Bellemare et al., 2017b), in practice, directly minimizing the Wasserstein distance is often not preferred as a loss function of a neural network and thus some of the exemplary works (Dabney et al., 2018b;a) of deep DRL minimizes the Huber quantile regression loss (Huber, 1964) instead. To this end, we treat the methods which minimize the Huber quantile loss as our baseline. However, as proven in Rowland

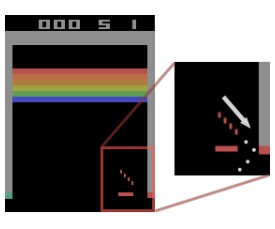

(a) State of interest

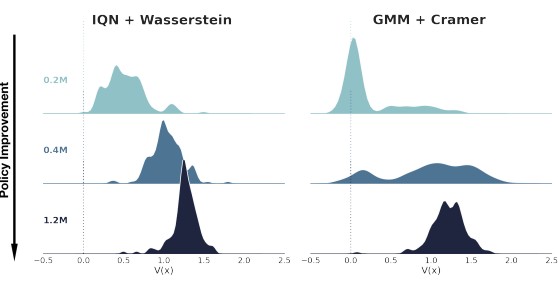

(b) The evaluated value distributions

Figure 1: Modality of value distribution during the learning process of Breakout-v4. (a) An arrow is added in the inset to indicate the ball's direction of travel. The episode reaches a terminal state if the paddle misses the ball. (b) Probability density functions of the value distributions learned by each actor-critic when $\{0.2, 0.4, 1.2\}$M frames are collected. As the policy improves, the probability of losing a turn ($V = 0$) decreases, and the probability of earning scores ($V > 0$) increases. Note that the modality transition from $V = 0$ is clearly captured by the GMM + Cramér method.

et al. (2019), representing a distribution using the Huber quantiles instead of samples can lead to conflation between statistics and samples, and thus an imputation strategy is required to learn a more accurate representation of a distribution. Despite its theoretical soundness, the imputation strategy can introduce computational overheads depending on the statistics (e.g. quantiles, expectiles). To avoid this issue, we formulate the DRL problem through samples and parameters instead of statistics, and minimize the Cramér distance between the value distributions. Combining these solutions, we arrive at a distributional actor-critic with Cramér distance as the value distribution loss.

On the other hand, many actor-critic methods suffer from data inefficiency and often include multi-step algorithms like the $\lambda$-return algorithm (Watkins, 1989) and off-policy updates, e.g. importance sampling. Here we address this problem by adapting multi-step off-policy updates to the distributional perspective, by defining a generalized form of the multi-step distributional Bellman targets. Furthermore, we introduce a novel value-distribution learning method which we call Sample-Replacement, denoted by SR($\lambda$). We show that the expectation of the target distribution from SR($\lambda$) is equivalent to the scalar $\lambda$-return. Additionally, we propose to parameterize the value distribution as a Gaussian mixture model (GMM). When combining GMM with the Cramér distance, we can derive an analytic solution and obtain unbiased sample gradients at a much lower computational cost compared to the method using Huber quantile loss. Altogether, we call our framework GMAC (Gaussian mixture actor-critic).

We present experimental results to demonstrate that this framework successfully outperforms the baseline algorithm with scalar value function in discrete action space, and can be expanded to continuous action spaces without any architectural or algorithmic modification. Furthermore, we show that more accurate representation of value distributions is learned, with a less computational cost.

## 2 RELATED WORKS

Bellemare et al. (2017a) has shown that the distributional Bellman operator derived from the distributional Bellman equation is a contractor in a maximal form of the Wasserstein distance. Based on this point, Bellemare et al. (2017a) proposed a categorical distributional model, C51, which is later discussed to be minimizing the Cramér distance in the projected distributional space (Rowland et al., 2018; Bellemare et al., 2019). Dabney et al. (2018b) proposed quantile regression-based models, QR-DQN, which parameterizes the distribution with a uniform mixture of Diracs and uses sample-based Huber quantile loss (Huber, 1964). Dabney et al. (2018a) later expand it further so that a full continuous quantile function can be learned through the implicit quantile network (IQN). Yang et al. (2019) then further improved the approximation of the distribution by adjusting the set of quantiles. Rowland et al. (2019) proposed expectile regression in place of quantile regression for learning categorical distribution to address the error in the Bellman target approximation. Choi et al. (2019) suggested parameterizing the value distribution using Gaussian mixture and minimizing the Tsallis-Jenson divergence as the loss function on a value-based method. Outside of RL, Bellemare

et al. (2017b) proposed to use Cramér distance in place of Wasserstein distance used in WGAN due to its unbiasedness in sample gradients (Arjovsky et al., 2017).

There have been many applications of the distributional perspective, which exploit the additional information from value distribution. Dearden et al. (1998) modeled parametric uncertainty and Morimura et al. (2010a;b) designed a risk-sensitive algorithm using a distributional perspective, which can be seen as the earliest concept of distributional RL. Mavrin et al. (2019) utilized the idea of the uncertainty captured from the variance of value distribution. Nikolov et al. (2019) has also utilized the distributional representation of the value function by using information-directed-sampling for better exploration of the value-based method. While multi-step Bellman target was considered (Hessel et al., 2018), the sample-efficiency was directly addressed by combining multi-step off-policy algorithms like Retrace($\lambda$) (Gruslys et al., 2017).

Just as C51 has been expanded deep RL to disitributional perspective, Barth-Maron et al. (2018) studied a distributional perspective on DDPG (Lillicrap et al., 2016), an actor-critic method, by parameterizing a distributional critic as categorical distribution and Gaussian mixture model. Singh et al. (2020) has further expanded the work by using an implicit quantile network for the critic. Several works (Duan et al., 2020; Kuznetsov et al., 2020; Ma et al., 2020) have proposed a distributional version of the soft-actor-critic (SAC) framework to address the error from over-estimating the value. However, previous works concentrated on extending a specific actor-critic framework to the distributional setting. Therefore, we aim to suggest methods which may be easily adopted in the process of expanding a scalar value methods to a distributional perspective, along with an attempt to address the previously mentioned issues present in the value-based ditributional algorithms.

## 3 DISTRIBUTIONAL REINFORCEMENT LEARNING

We consider a conventional RL setting, where an agent's interaction with its environment is described by a Markov Decision Process (MDP) $(\mathcal{X}, \mathcal{A}, R, P, \gamma)$, where $\mathcal{X}$ and $\mathcal{A}$ are state and action spaces, $R(x, a)$ is the stochastic reward function for a pair of state $x$ and action $a$, $P(x'|x, a)$ is the transition probability of observing $x'$ given the pair $(x, a)$, and $\gamma \in (0,1)$ is a time discount factor. A policy $\pi(\cdot|x)$ maps a state $x \in \mathcal{X}$ to a probability distribution over actions $a \in \mathcal{A}$.

The objective of RL is to maximize the expected return, $\mathbb{E}[G_t]$ where $G_t = \sum_{t=0}^{\infty} \gamma^t R(x_t, a_t)$ is the sum of discounted rewards from state $x_t$ given a policy $\pi$ at time $t$. Then for any state $x_t$, the value $V$ and state-action value $Q$ under the given policy $\pi$ can be defined as

$$V(x_t) = \mathbb{E}[G_t \mid X = x_t], \quad Q(x_t, a_t) = \mathbb{E}[G_t \mid X = x_t, A = a_t]. \tag{1}$$

A recursive relationship in the value in terms of the reward and the random transition is described by the Bellman equation (Bellman, 1957) given by

$$Q(x, a) = \mathbb{E}[R(x, a)] + \gamma \mathbb{E}_{a' \sim \pi, x' \sim P} [Q(x', a')], \tag{2}$$

where the first expectation is calculated over a given state-action pair $(x, a)$ and the second expectation is taken over the next possible states $x' \sim P(\cdot|x, a)$ and actions $a' \sim \pi(\cdot|x)$.

DRL extends the Bellman equation to an analogous recursive equation, termed the distributional Bellman equation (Morimura et al., 2010a;b; Bellemare et al., 2017a), using a distribution of the possible sum of discounted rewards $Z(x, a)$:

$$Z(x, a) \stackrel{D}{=} R(x, a) + \gamma Z(x', a'), \tag{3}$$

where $\stackrel{D}{=}$ denotes having equal distributions and $Q(x, a) = \mathbb{E}[Z(x, a)]$. Then $Z$ is learned through distributional Bellman operator $\mathcal{T}^\pi$ defined as

$$\mathcal{T}^\pi Z(x, a) \stackrel{D}{:=} R(x, a) + \gamma P^\pi Z(x, a) \tag{4}$$

where $P^\pi : \mathcal{Z} \to \mathcal{Z}$ is a state transition operator under policy $\pi$, $P^\pi Z(x, a) \stackrel{D}{:=} Z(x', a')$. Analogously, the distributional Bellman optimality operator $\mathcal{T}$ can be defined as

$$\mathcal{T} Z(x, a) \stackrel{D}{:=} R(x, a) + \gamma Z(x', \arg\max_{a'} \mathbb{E}_{x' \sim P}[Z(x', a')]). \tag{5}$$

---

**Algorithm 1:** SR($\lambda$)

---

**Input:** Trajectory of states and values $\{(x_1, Z_1), \ldots, (x_T, Z_T)\}$ for a given length $T$, discount factor $\gamma$, weight parameter $\lambda$

**Output:** Set of $\lambda$-returns $\{Z_1^{(\lambda)}, \ldots, Z_{T-1}^{(\lambda)}\}$

$\boldsymbol{X} \leftarrow$ Collect $m$ samples $\{X_1, \ldots, X_m\}$ from $Z_T$

**for** $t = T - 1$ **to** 1 **do**

    $\boldsymbol{X} \leftarrow r_t + \gamma \boldsymbol{X}$ // Bellman operator

    $Z_t^{(\lambda)} \leftarrow \sum_{i=1}^{m} \delta_{X_i}$ // empirical distribution using $m$ Diracs

    $\boldsymbol{X}' \leftarrow$ Collect $m$ samples $\{X_1', \ldots, X_m'\}$ from $Z_t^{(\lambda)}$

    **for** $i = 1$ **to** $m$ **do**

        $X_i \leftarrow X_i'$ with probability $1 - \lambda$

    **end**

**end**

---

The distributional Bellman operator has been proven to be a $\gamma$-contraction in a maximal form of Wasserstein distance (Bellemare et al., 2017a), which has a practical definition given by

$$d_p(U, V) = \left( \int_0^1 |F_U^{-1}(\omega) - F_V^{-1}(\omega)|^p d\omega \right)^{1/p}, \tag{6}$$

where $U, V$ are random variables and $F_U, F_V$ are their cumulative distribution functions (cdf).

However, unlike the distributional Bellman operator, the distributional Bellman optimality operator is not a contractor in any metric (Bellemare et al., 2017a). Thus, the distance $d_p(\mathcal{T}Z_1, \mathcal{T}Z_2)$ between some random variables $Z_1, Z_2$ may not converge to a unique solution. This issue has been discussed in Bellemare et al. (2017a), with an example of oscillating value distribution caused by a specific tie-breaker design of the *argmax* operator. One way to remove this issue from consideration is by learning the value distributions via expected Bellman operator with a trainable stochastic policy and finding an optimal policy under principles of conservative policy iteration by Kakade & Langford (2002). See Appendix A for more discussion.

## 4 ALGORITHM

In this section, we incrementally develop the building blocks of our proposed method. First, we present an efficient distributional version of $\lambda$-return algorithm called Sample-Replacement, denoted by SR($\lambda$). Then, we show that minimizing the energy distance, which is equivalent to a specific form of the Cramér distance, between Gaussian mixtures can be a better solution than using quantile regressions when working with distributional actor-critic and SR($\lambda$).

### 4.1 SR($\lambda$): SAMPLE-REPLACEMENT FOR $\lambda$-RETURN DISTRIBUTION

The actor-critic method is a temporal-difference (TD) learning method in which the value function, the critic, is learned through the TD error defined by the difference between the TD target given by $n$-step return, $G_t^{(n)} = \sum_{i=1}^{n} \gamma^{i-1} r_{t+i} + \gamma^n V(x_{t+n})$, and the current value estimate $V(x_t)$. A special case of TD method, called TD($\lambda$) (Sutton, 1988), generates a weighted average of $n$-step returns for the TD target, also known as the $\lambda$-return,

$$G_t^{(\lambda)} = (1 - \lambda) \sum_{n=1}^{\infty} \lambda^{n-1} G_t^{(n)}, \quad \lambda \in [0, 1], \tag{7}$$

to mitigate the variance and bias trade-off between Monte Carlo and the TD(0) return to enhance data efficiency. From an alternative perspective, equation 7 can be thought of as finding a TD target via taking the expectation of a random variable $\tilde{G}$ whose sample space is the set of all $n$-step returns, $\{G_t^{(0)}, \ldots, G_t^{(\infty)}\}$. Then the probability distribution of $\tilde{G}$ is given by

$$\Pr[\tilde{G} = G_t^{(n)}] = (1 - \lambda)\lambda^{n-1}. \tag{8}$$

Similar to $G_t^{(n)}$, we define $n$-step approximation of the value distribution as

$$Z_t^{(n)} \stackrel{D}{:=} \sum_{i=0}^{n-1} \gamma^i R(x_{t+i}, a_{t+i}) + \gamma^n \mathbb{E}_{a' \sim \pi}[Z(x_{t+n}, a')], \tag{9}$$

where $\mathbb{E}[Z_t^{(n)}] = G_t^{(n)}$. Then, a distributional analogy of equation 8 can be written as

$$\Pr[\tilde{Z} = Z_t^{(n)}] = (1 - \lambda)\lambda^{n-1} \tag{10}$$

where $\tilde{Z}$ is a random variable whose sample space is a set of all $n$-step approximations, $\{Z_t^{(0)}, \ldots, Z_t^{(\infty)}\}$. Unlike $\tilde{G}$, we cannot directly calculate an expectation over a set of random variables, $\tilde{Z}$. Instead, we redefine equation 10 in terms of cdfs:

$$\Pr[\tilde{F} = F_{Z_t^{(n)}}] = (1 - \lambda)\lambda^{n-1}. \tag{11}$$

$F_{Z_t^{(n)}}$ denotes the cdf of the $n$-step return $Z_t^{(n)}$, and $\tilde{F} = \{F_{Z_t^{(0)}}, \ldots, F_{Z_t^{(\infty)}}\}$ is a random variable over the set of $F_{Z_t^{(n)}}$. Then, for any $z$, we can successfully rewrite equation 11 as a linear combination of $F_{Z_t^{(n)}}$

$$\mathbb{E}[\tilde{F}] = (1 - \lambda) \sum_{n=1}^{\infty} \lambda^{n-1} F_{Z_t^{(n)}}. \tag{12}$$

Let us define a random variable $Z_t^{(\lambda)}$ that has $\mathbb{E}[\tilde{F}]$ as its cdf. Then the expectation of $Z_t^{(\lambda)}$ and the expectation of $Z_t^{(n)}$ have an analogous relationship to equation 7 (see Appendix B), meaning that its behavior in expectation is equal to that of the $\lambda$-return. Therefore, we treat the resulting random variable $Z_t^{(\lambda)}$ as a distributional analogy of the $\lambda$-return. We note that, in practice, collecting infinite horizon trajectory is infeasible and thus we use a truncated sum (Cichosz, 1995; van Seijen et al., 2011):

$$F_{Z_t^{(\lambda)}} = (1 - \lambda) \sum_{n=1}^{N} \lambda^{n-1} F_{Z_t^{(n)}} + \lambda^N F_{Z_t^{(N)}}. \tag{13}$$

Given a trajectory of length $N$, naïvely speaking, finding $Z_t^{(\lambda)}$ for each time step requires finding $N$ different $Z_t^{(n)}$. As a result, we need to find total of $O(N^2)$ different distributions to find $Z_t(\lambda)$ for all states in the given trajectory. The number of distributions to find can be reduced to $O(N)$, in practice, by approximating the distribution of $Z_t^{(n)}$ with a mixture of diracs, as described in Dabney et al. (2018b), because the Bellman operations can be applied to the same set of samples to find $Z_t^{(n)}$ for different $t$'s. Then, the approximation takes the form of

$$Z_t^{(n)} \approx Z_\theta(x_t) := \frac{1}{m} \sum_{i=1}^{m} \delta_{\theta_i(x_t)} \tag{14}$$

where $\theta : \mathcal{X} \to \mathbb{R}^m$ is some parametric model. To obtain the target samples for $Z_\theta(x_t)$, we start with $m$ samples of the last value distribution in the sampled trajectory. Traversing through the sampled trajectory in a reversed order, we replace each sample with a new sample from the next state with a probability of $1 - \lambda$. Then the obtained set $\mathbf{X}_m$ is equivalent to a set of samples from the approximated distribution of the $\lambda$-returns, $Z_t^{(\lambda)}$. A more detailed description of the algorithm can be found in Algorithm 1.

For sample-based methods like the implicit quantile network, one can directly use this set $\mathbf{X}_m$ as the target samples. However, Rowland et al. (2019) has shown that the quantiles predicted from the Huber quantile loss cannot be interpreted as samples, thus an imputation strategy is required to generate a distribution from the statistics. We propose to minimize the Cramér distance instead in which the predicted parameters can be samples or the parameters of the distribution itself.

### 4.2 CRAMÉR DISTANCE

Let $P$ and $Q$ be probability distributions over $\mathbb{R}$. If we define the cdf of $P, Q$ as $F_P, F_Q$ respectively, the $l_p$ family of divergence between $P$ and $Q$ is

$$l_p(P, Q) := \left( \int_{-\infty}^{\infty} |F_P(x) - F_Q(x)|^p dx \right)^{1/p}. \tag{15}$$

When $p = 2$, it is termed the Cramér distance. The distributional Bellman operator in the evaluation setting is a $|\gamma|^{1/p}$-contraction mapping in the Cramér metric space (Rowland et al., 2019), whose worked out proof can be found in Appendix C.

A notable characteristic of the Cramér distance is the unbiasedness of the sample gradient,

$$\mathop{\mathbb{E}}_{X \sim Q} \nabla_\theta l_2^2(\hat{P}_m, Q_\theta) = \nabla_\theta l_2^2(P, Q_\theta) \tag{16}$$

where $\boldsymbol{X}_m := \{X_1, ..., X_m\}$ are samples drawn from $P$, $\hat{P}_m := \frac{1}{m} \sum_{i=1}^m \delta_{X_i}$ is the empirical distribution, and $Q_\theta$ is a parametric approximation of a distribution. The unbiased sample gradient makes it suitable to use Cramér distance with stochastic gradient descent method and empirical distributions for updating the value distribution.

Székely (2002) showed that, in the univariate case, the squared Cramér distance is equivalent to one half of *energy distance* ($l_2^2(P, Q) = \frac{1}{2}\mathcal{E}(P, Q)$) defined as

$$\mathcal{E}(P, Q) := \mathcal{E}(U, V) = 2\,\mathbb{E}\,\|U - V\|_2 - \mathbb{E}\,\|U - U'\|_2 - \mathbb{E}\,\|V - V'\|_2, \tag{17}$$

where $U, U'$ and $V, V'$ are random variables that follow $P, Q$, respectively.

### 4.3 ENERGY DISTANCE BETWEEN GAUSSIAN MIXTURE MODELS

So far, we have described the components required to formulate a general setting of the suggested distributional extension to actor-critic methods. Here, we take one step further to enhance the approximation accuracy and computational efficiency by considering the parameterized model of the value distribution as a Gaussian mixture model (Choi et al., 2019; Barth-Maron et al., 2018). Following the same assumption used for equation 14 the approximation using Gaussian mixture is given using parametric models $\mu, \sigma, w : \mathcal{X} \to \mathbb{R}^K$

$$Z_\theta(x_t) := \sum_{i=1}^K w_i(x_t)\,\mathcal{N}(z; \mu_i(x_t), \sigma_i(x_t)^2), \text{ where } \sum_{i=1}^K w_i(x_t) = 1. \tag{18}$$

If random variables $X, Y$ follow the distributions $P, Q$ parameterized as GMM, the energy distance has the following closed-form

$$\mathcal{E}(X, Y) = 2\delta(X, Y) - \delta(X, X') - \delta(Y, Y'),$$
$$\text{where } \delta(U, V) = \sum_{i,j} w_{ui} w_{vj}\, \mathbb{E}\left[ \left| \mathcal{N}(z; \mu_{ui} - \mu_{vj}, \sigma_{ui}^2 + \sigma_{vj}^2) \right| \right]. \tag{19}$$

Here, $\mu_{xi}$ refers to the $i^{th}$ component for random variable $X$ and same applies for $\sigma$ and $w$. With Gaussian mixtures, the closed-form solution of the energy distance defined in equation 19 has a computational advantage over sample-based approximations like the Huber quantile loss. When using GMM, the analytic approximation of equation 14 can be derived as

$$Z_t^{(\lambda)} \approx \sum_{n=1}^{\infty} (1-\lambda)\lambda^{n-1} \sum_{k=1}^K w_{nk}\,\mathcal{N}(z; \mu_{nk}, \sigma_{nk}^2) \approx \frac{1}{m} \sum_{i=1}^m \mathcal{N}(z; \mu_{nk}, \sigma_{nk}^2), \tag{20}$$

where $\mu_{nk}$ refers to $k^{th}$ component of $\mu_{Z_t^{(n)}}$ for simplicity of notation, $n$ is sampled from $\text{Geo}(1-\lambda)$ and the index $k \in \{1, \ldots, K\}$ is sampled proportional to the mixture weights $\{w_1, \ldots, w_K\}$. This is equivalent to having a mixture of $m$ Gaussians, thus we can simply perform sample replacement on the parameters $(\mu, \sigma^2)$, instead of realizations of the random variables as in equation 14. Then, the loss function described in equation 19 can easily be applied.

When bringing all the components together, we have a distributional actor-critic framework with SR($\lambda$) that minimizes the Cramér distance between Gaussian mixture value distributions. We call this method GMAC. A brief sketch of the algorithm is shown in Appendix E.

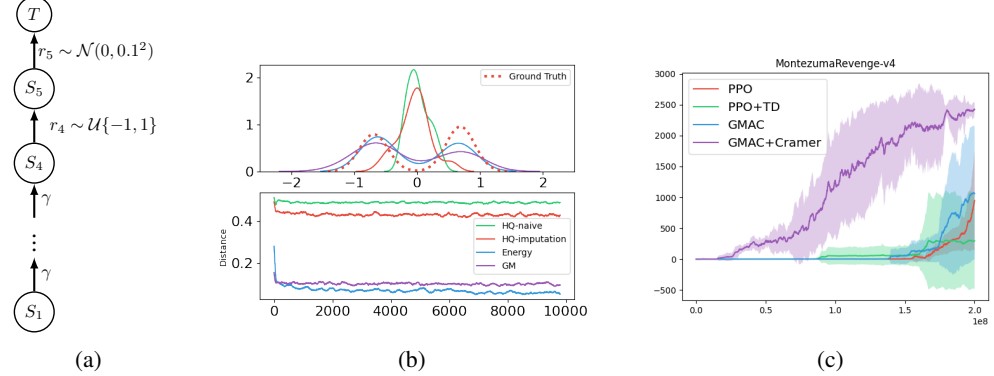

Figure 2: (a) An environment with two states and stochastic rewards with expected value of zero. (b) The probability density function (above) and the Cramér distance between the ground truth and the estimated distributions for IQ, IQE, and GM (below). (c) Learning curve of Montezuma's Revenge using modality information as intrinsic reward.

## 5 EXPERIMENTS

In this section, we present experimental results for three different distributional versions of Proximal Policy Optimization (PPO) with SR($\lambda$): IQAC (IQN + Huber quantile), IQAC-E (IQN + energy distance), and GMAC (GMM + energy distance), in the order of the progression of our suggested approach. The performance of the scalar version of PPO with value clipping (Schulman et al., 2016) is used as the baseline for comparison. Details about the loss function of each method can be found in Appendix D. For a fair comparison, we keep all common hyperparameters consistent across the algorithms except for the value heads and their respective hyperparameters (see Appendix F).

The results demonstrate three contributions of our proposed DRL framework: 1) the ability to correctly capture the multimodality of value distributions, 2) generalization to both discrete and continuous action spaces, and 3) significantly reduced computational cost.

**Representing Multimodality** As discussed throughout Section 4, we expect minimizing the Cramér distance to produce a more appropriate depiction of a distribution compared to minimizing the Huber quantile loss. First, we demonstrate this with a simple value regression problem for an MDP of five sequential states, as shown in Figure 2 (a). The reward function $r_i$ of last two state $S_i$ is stochastic, with $r_4$ from a uniform discrete distribution and $r_5$ from a normal distribution. Then the value distribution of $S_1$ should be bimodal with expectation of zero (Figure 2 (b)). In this example, minimizing the Huber quantile loss (labeled as IQ-naive) of a implicit quantile network underestimates the variance of $S_1$ due to conflation and does not capture the locations of the modes. Applying an imputation strategy as suggested in Rowland et al. (2019), improvement on the underestimation of variance can be seen. On the other hand, minimizing the Cramér distance converges to correct mode locations using both implicit quantile network and Gaussian mixture model, labeled as IQE and GM respectively in the figure. More details about the experimental setup and further results can be found in Appendix G. The comparison can be extended to more complex tasks such as the Atari games, of which a sample result is shown in Figure 1, and additional visualizations of the value distribution during the learning process from different games can be found in Appendix G.

So what can be achieved with correct modality? By capturing the correct modes of the distribution, an additional degree of freedom on top of the expected value can be accurately obtained, from which richer information can be derived to distinguish states by their value distributions. In particular, the extra information may be utilized as an intrinsic motivation in sparse-reward exploration tasks. To demonstrate, we compare using Cramér distance between value distributions as intrinsic reward to using TD error between scalar value estimates in a sparse reward environment of Montezuma's Revenge in Figure 2 (c), which shows a clear improvement in the performance.

**Discrete and Continuous Action Spaces** Experimental results from the ALE (Bellemare et al., 2013) and the PyBullet environments (Coumans & Bai, 2016–2020) show that our algorithm can be

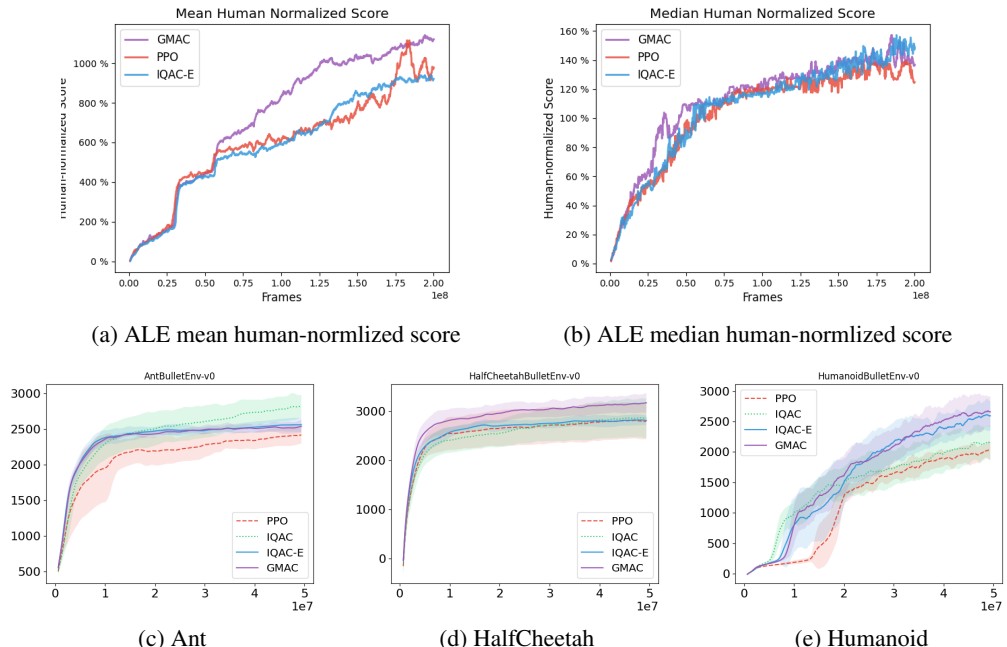

Figure 3: Learning curves for Atari games from ALE and 3 continuous control tasks from PyBullet.

applied to both discrete and continuous action spaces. While GMAC shows overall improvement in 61 Atari games (Appendix G), we present the full learning curves of 3 Atari games in Figure 3 in which previous distributional methods have shown improvements over the scalar version. One can observe that the results from GMAC show the most significant improvement from the baseline.

All experiments are run over 5 random seeds with consistent hyperparameters. In Figure 3, a solid line represents an average of mean scores over 100 recent episodes and the shaded area represents the standard deviation over the seeds. For visual clarity, the plots are smoothed over 5M frames and 1.25M frames in Atari and PyBullet, respectively. Non-smoothed learning curves on more tasks and the final scores over the 61 ALE tasks can be found in Appendix G.

**Computational Cost** Table 1 shows the number of parameters and the number of floating-point operations (FLOPs) required for a single inference and update step of each agent. We emphasize three points here. First, the implicit quantile network requires more parameters due to the intermediate embeddings of random quantiles. Second, the difference between the FLOPs for a single update in IQAC and IQAC-E indicates that the proposed energy distance requires less computation than the Huber quantile regression. Last, the results for GMAC show that using Gaussian Mixtures can greatly reduce the cost even to match the numbers of PPO while having improved performance.

Table 1: FLOP measurement results for a single process in Breakout-v4

| Algorithm | Params (M) | FLOPs (G) | |
| --- | --- | --- | --- |
| | | **Inference** | **Update** |
| PPO | 0.44 | 1.73 | 5.19 |
| IQAC | 0.52 | 2.98 | 12.98 |
| IQAC-E | 0.52 | 2.98 | 8.98 |
| GMAC | **0.44** | **1.73** | **5.27** |

## 6 CONCLUSION

In this paper, we have developed the distributional perspective of the actor-critic framework which integrates the SR($\lambda$) method, Cramér distance, and Gaussian mixture models for improved performance in both discrete and continuous action spaces at a lower computational cost. Furthermore, we show that our proposed method can capture the correct modality in the value distribution, while the extension of the conventional method with the stochastic policy fails to do so.

Capturing the correct modality of value distributions can improve the performance of various policy-based RL applications that exploit statistics from the value distribution. Such applications may include training risk-sensitive policies and learning control tasks with sparse rewards that require heavy exploration, where transient information from the value distribution can give benefit to the learning process. We leave further development of these ideas as future works.

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

# Appendices

## A   DISCUSSION ON THE CHOICE OF PROXIMAL POLICY OPTIMIZATION AS A BASELINE

A general learning process of RL can be described using policy iteration, which consists of two iterative phases: policy evaluation and policy improvement (Sutton & Barto, 1998). In policy iteration, the value function is assumed to be exact, meaning that given policy, the value function is learned until convergence for the entire state space, which results in a strong bound on the rate of convergence to the optimal value and policy (Puterman, 1994).

But the exact value method is often infeasible from resource limitation since it requires multiple sweeps over the entire state space. Therefore, in practice, the value function is approximated, i.e. it is not trained until convergence nor across the entire state space on each iteration. The approximate version of the exact value function method, also known as *asynchronous value iteration*, still converges to the unique optimal solution of the Bellman optimality operator. However, the Bellman optimality only describes the limit convergence, and thus the best we can practically consider is to measure the improvement on each update step.

Bertsekas & Tsitsiklis (1996) have shown that, when we approximate the value function $V_\pi$ of some policy $\pi$ with $\tilde{V}$, the lower bound of a greedy policy $\pi'$ is given by

$$V_{\pi'}(x) \geq V_\pi(x) - \frac{2\gamma\varepsilon}{1-\gamma}, \tag{21}$$

where $\varepsilon = \max_x |\tilde{V}(x) - V_\pi(x)|$ is the $L_\infty$ error of value approximation $\tilde{V}$. This means a greedy policy from an approximate value function guarantees that its exact value function will not degrade more than $\frac{2\gamma\varepsilon}{1-\gamma}$. However, there is no guarantee on the improvement, i.e. $V_{\pi'}(x) > V_\pi(x)$ (Kakade & Langford, 2002).

As a solution to this issue, Kakade & Langford (2002) have proposed a policy updating scheme named *conservative policy iteration*,

$$\pi_{new}(a|x) = (1-\alpha)\pi_{old}(a|x) + \alpha\pi'(a|x), \tag{22}$$

which has an explicit lower bound on the improvement

$$\eta(\pi_{new}) \geq L_{\pi_{old}}(\pi_{new}) - \frac{2\epsilon\gamma}{(1-\gamma)^2}\alpha^2, \tag{23}$$

where $\epsilon = \max_x |\mathbb{E}_{\pi'}[A_\pi(x,a)]|$, $A_\pi(x,a) = Q(x,a) - V(x)$ is the advantage function, $\eta(\pi)$ denotes the expected sum of reward under the policy $\pi$,

$$\eta(\pi) = \mathbb{E}\left[\sum_{t=0}^{\infty} \gamma^t R(x_t, a_t)\right], \tag{24}$$

and $L_{\pi_{old}}$ is the local approximation of $\eta$ with the state visitation frequency under the old policy.

From the definition of distributional Bellman optimality operator in equation 5, one can see that the lower bound in equation 23 also holds when $\pi'$ is greedy with respect to the expectation of the value distribution, i.e., $\mathbb{E}_{x'\sim P}[Z(x',a')]$. Thus the improvement of the distributional Bellman update is guaranteed in expectation under conservative policy iteration, and the value functions are guaranteed to converge in distribution to a fixed point by $\gamma$-contraction.

Schulman et al. (2015) takes this further, suggesting an algorithm called trust region policy optimization (TRPO), which extends conservative policy iteration to a general stochastic policy by replacing $\alpha$ with Kullback-Leibler (KL) divergence between two policies,

$$D_{KL}^{max}(\pi, \tilde{\pi}) = \max_x D_{KL}\left(\pi(\cdot|x)\|\tilde{\pi}(\cdot|x)\right). \tag{25}$$

Then, the newly formed objective is to maximize the following, which is a form of constraint optimization with penalty:

$$\hat{\mathbb{E}}_t\left[\frac{\pi(a_t|x_t)}{\tilde{\pi}(a_t|x_t)}\hat{A}_t - \beta D_{KL}(\pi(\cdot|x_t), \tilde{\pi}(\cdot|x_t))\right] = \hat{\mathbb{E}}_t\left[r_t(\pi)\hat{A}_t - \beta D_{KL}(\pi(\cdot|x_t), \tilde{\pi}(\cdot|x_t))\right]. \tag{26}$$

where $r(\pi)$ refers to the ratio $r(\pi) = \frac{\pi(a_t|x_t)}{\tilde{\pi}(a_t|x_t)}$. However, in practice, choosing a fixed penalty coefficient $\beta$ is difficult and thus Schulman et al. (2015) uses hard constraint instead of the penalty.

$$\max_\theta \hat{\mathbb{E}}_t \left[ r_t(\pi)\hat{A}_t \right] \tag{27}$$

$$\text{s.t. } D_{KL}(\pi(\cdot|x_t), \tilde{\pi}(\cdot|x_t)) \le \delta \tag{28}$$

Schulman et al. (2017) simplifies the loss function even further in proximal policy optimization (PPO) by replacing KL divergence with ratio clipping between the old and the new policy with the following:

$$L^{CLIP} = \hat{\mathbb{E}}_t \left[ \min \left( r_t(\pi)\hat{A}_t, \text{clip}(r_t(\pi), 1 - \epsilon, 1 + \epsilon)\hat{A}_t \right) \right]. \tag{29}$$

Thus, by using PPO as the baseline, we aim to optimize the value function via unique point convergence of distributional Bellman operator for a policy being approximately updated under the principle of conservative policy.

## B  EXPECTATION VALUE OF $Z_t^{(\lambda)}$

Continuing from equation 12, let us define a random variable that has a cumulative distribution function of $\mathbb{E}[\tilde{F}_Z]$ as $Z_t^{(\lambda)}$. Then, its cumulative distribution function is given by

$$F_{Z_t^{(\lambda)}} = (1 - \lambda) \sum_{n=1}^{\infty} \lambda^{n-1} F_{Z_t^{(n)}}. \tag{30}$$

If we assume that the support of $Z_t^{(\lambda)}$ is defined in the extended real line $[-\infty, \infty]$,

$$\mathbb{E}[Z_t^{(\lambda)}] = \int_0^\infty \left( 1 - F_{Z_t^{(\lambda)}} \right) dz - \int_{-\infty}^0 F_{Z_t^{(\lambda)}} dz \tag{31}$$

$$= \int_0^\infty \left( 1 - (1 - \lambda) \sum_{n=1}^{\infty} \lambda^{n-1} F_{Z_t^{(n)}} \right) dz - \int_{-\infty}^0 (1 - \lambda) \sum_{n=1}^{\infty} \lambda^{n-1} F_{Z_t^{(n)}} dz \tag{32}$$

$$= (1 - \lambda) \sum_{n=1}^{\infty} \lambda^{n-1} \left[ \int_0^\infty \left( 1 - F_{Z_t^{(n)}} \right) dz - \int_{-\infty}^0 F_{Z_t^{(n)}} dz \right] \tag{33}$$

$$= (1 - \lambda) \sum_{n=1}^{\infty} \lambda^{n-1} G_t^{(n)} = G_t^{(\lambda)}. \tag{34}$$

Thus we can arrive at the desired expression of $\mathbb{E}[Z_t^{(\lambda)}] = G_t^{(\lambda)}$.

## C  DISTRIBUTIONAL BELLMAN OPERATOR AS A CONTRACTION IN CRAMÉR METRIC SPACE

The Cramér distance possesses the following characteristics (detailed derivation of each can be found in (Bellemare et al., 2017b)):

$$l_p(A + X, A + Y) \le l_p(X, Y), \qquad l_p(cX, cY) \le |c|^{1/p} l_p(X, Y). \tag{35}$$

Using the above characteristics, the Bellman operator in $l_p$ divergence is

$$\begin{aligned} l_p \left( \mathcal{T}^\pi Z_1(x, a), \mathcal{T}^\pi Z_2(x, a) \right) &= l_p(R(x, a) + \gamma P^\pi Z_1(x, a), R(x, a) + \gamma P^\pi Z_2(x, a)) \\ &\le |\gamma|^{1/p} l_p(P^\pi Z_1(x, a), P^\pi Z_2(x, a)) \\ &\le |\gamma|^{1/p} \sup_{x', a'} l_p(Z_1(x', a'), Z_2(x', a')). \end{aligned} \tag{36}$$

Substituting the result into the definition of the maximal form of the Cramér distance yields

$$
\begin{aligned}
\bar{l}_p(\mathcal{T}^\pi Z_1, \mathcal{T}^\pi Z_2) &= \sup_{x,a} l_p(\mathcal{T}^\pi Z_1(x,a), \mathcal{T}^\pi Z_2(x,a)) \\
&\leq |\gamma|^{1/p} \sup_{x',a'} l_p(Z_1(x',a'), Z_2(x',a')) \\
&= |\gamma|^{1/p} \bar{l}_p(Z_1, Z_2).
\end{aligned}
\tag{37}
$$

Thus the distributional Bellman operator is a $|\gamma|^{1/p}$-contraction mapping in the Cramér metric space, which was also proven in Rowland et al. (2019).

Similar characteristics as in equation 35 can be derived for the energy distance

$$
\mathcal{E}(A+X, A+Y) \leq \mathcal{E}(X,Y), \qquad\qquad \mathcal{E}(cX, cY) = c\mathcal{E}(X,Y),
\tag{38}
$$

showing that the distributional Bellman operator is a $\gamma$-contractor in energy distance

$$
\mathcal{E}(\mathcal{T}^\pi Z_1, \mathcal{T}^\pi Z_2) \leq \gamma \mathcal{E}(Z_1, Z_2).
\tag{39}
$$

# D    LOSS FUNCTIONS

As in other policy gradient methods, our value distribution approximator models the distribution of the value, $V(x_t)$, not the state-action value $Q(x_t, a_t)$, and denote it as $Z_\theta(x_t)$ parametrized with $\theta$, whose cumulative distribution function is defined as

$$
F_{Z_\theta(x_t)} = \sum_{a \in \mathcal{A}} \pi(a, x_t) F_{Z(x_t, a)}.
\tag{40}
$$

Below, we provide the complete loss function of value distribution approximation for each of the cases used in experiments (Section 5).

## D.1    IMPLICIT QUANTILE + HUBER QUANTILE (IQAC)

For the value loss of IQAC, we follow the general flow of Huber quantile loss described in Dabney et al. (2018b). For two random samples $\tau, \tau' \sim U([0,1])$,

$$
\delta_t^{\tau, \tau'} = Z_t^{(\lambda)}(x_t, a_t; \tau') - Z_\theta(x_t; \tau)
\tag{41}
$$

where $Z_t^{(\lambda)}$ is generated via SR($\lambda$) and $Z(x; \tau) = F_Z^{-1}(\tau)$ is realization of $Z(X)$ given $X = x$ and $\tau$. Then, the full loss function of value distribution is given by

$$
L_{Z_\theta} = \frac{1}{N'} \sum_{i=1}^{N} \sum_{j=1}^{N'} \rho_{\tau_i}^\kappa \left( \delta_t^{\tau_i, \tau_j'} \right)
\tag{42}
$$

where $N$ and $N'$ are number of samples of $\tau, \tau'$, respectively, and $\rho$ is the Huber quantile loss

$$
\rho_\tau^\kappa(\delta_{ij}) = |\tau - \mathbb{I}\{\delta_{ij} < 0\}| \frac{L_\kappa(\delta_{ij})}{\kappa}, \quad \text{with}
\tag{43}
$$

$$
L_\kappa(\delta_{ij}) = \begin{cases} \frac{1}{2}\delta_{ij}^2, & \text{if} |\delta_{ij}| \leq \kappa \\ \kappa(|\delta_{ij}| - \frac{1}{2}\kappa), & \text{otherwise.} \end{cases}
\tag{44}
$$

## D.2    IMPLICIT QUANTILE + ENERGY DISTANCE (IQAC-E)

Here, we replace the Huber quantile loss in equation 42 with sample-based approximation of energy distance defined in equation 19.

$$
L_{Z_\theta} = \frac{2}{NN'} \sum_{i=1}^{N} \sum_{j=1}^{N'} \left| \delta_t^{\tau_i, \tau_j'} \right| - \frac{1}{N^2} \sum_{i=1}^{N} \sum_{i'=1}^{N} \left| \delta_t^{\tau_i, \tau_{i'}} \right| - \frac{1}{N'^2} \sum_{j=1}^{N'} \sum_{j'=1}^{N'} \left| \delta_t^{\tau_{j'}', \tau_j'} \right|
\tag{45}
$$

### D.3 GAUSSIAN MIXTURE + ENERGY DISTANCE (GMAC)

Unlike the two previous losses, which use samples at $\tau$ generated by the implicit quantile network $Z_\theta(x_t; \tau)$, here we discuss a case in which the distribution is $k$-component Gaussian mixture parameterized with $(\mu_k, \sigma_k^2, w_k)$.

Using the expectation of a folded normal distribution, we define $\delta$ between two Gaussian distributions as

$$\delta(\mu_i, \sigma_i^2, \mu_j, \sigma_j^2) = \sqrt{\frac{2}{\pi}} \sqrt{\sigma_i^2 + \sigma_j^2} \exp\left(-\frac{(\mu_i - \mu_j)^2}{2(\sigma_i^2 + \sigma_j^2)}\right) + (\mu_i - \mu_j)\left[1 - 2\Phi\left(\frac{(\mu_i - \mu_j)}{\sqrt{2}}\right)\right].$$

$$(46)$$

Let $Z_\theta(x)$ and $Z_t^{(\lambda)}$ be Gaussian mixtures parameterized with $(\mu_{\theta i}, \sigma_{\theta i}^2, w_{\theta i}), (\mu_{\lambda j}, \sigma_{\lambda j}^2, w_{\lambda j})$, respectively. Then, the loss function for the value head is given by

$$\begin{aligned}
L_{Z_\theta} = &\frac{2}{NN'} \sum_{i=1}^{N} \sum_{j=1}^{N'} w_{\theta i} w_{\lambda j} \delta(\mu_{\theta i}, \sigma_{\theta i}^2, \mu_{\lambda j}, \sigma_{\lambda j}^2) \\
&-\frac{1}{N^2} \sum_{i=1}^{N} \sum_{i'=1}^{N} w_{\theta i} w_{\theta i'} \delta(\mu_{\theta i}, \sigma_{\theta i}^2, \mu_{\theta i'}, \sigma_{\theta i'}^2) \\
&-\frac{1}{N'^2} \sum_{j=1}^{N'} \sum_{j'=1}^{N'} w_{\lambda j} w_{\lambda j'} \delta(\mu_{\lambda j}, \sigma_{\lambda j}^2, \mu_{\lambda j'}, \sigma_{\lambda j'}^2).
\end{aligned}$$

$$(47)$$

# E    PSEUDOCODE OF GMAC

---

**Algorithm 2:** GMAC

---

**Input:** Initial policy parameters $\theta_0$, initial value function parameters $\phi_0$, length of trajectory $T$,
number of environments $E$, clipping factor $\epsilon$, discount factor $\gamma$, weight parameter $\lambda$

**for** $k = 0, 1, 2, \ldots$ **do**

    **for** $e = 1, \ldots, E$ **do**

        Collect samples of discounted sum of rewards $\{Z_1, \ldots, Z_T\}$ by running policy
$\pi_k = \pi(\theta_k)$ in the environment

        Compute the parameters $(\mu_i, \sigma_i, w_i)$ for each of the $\lambda$-returns $\{Z_1^{(\lambda)}, \ldots, Z_{T-1}^{(\lambda)}\}$ by
SR($\lambda$) (Algorithm 1)

        Compute advantage estimates $\hat{A}_t$ using GAE (Schulman et al., 2016), based on the
current value function $V_{\phi_k}$

    **end**

    Gather the data from $E$ environments

    Update policy using the clipped surrogate loss:

$$\theta_{k+1} = \arg\max_{\theta} \mathbb{E}\left[\min\left(\frac{\pi_\theta(a_t|s_t)}{\pi_{\theta_k}(a_t|s_t)}\hat{A}_t,\ g(\epsilon, \hat{A}_t)\right)\right]$$

    via stochastic gradient ascent.

    Update value function using the energy distance between Gaussian mixtures (Equation 19):

$$\phi_{k+1} = \arg\min_{\phi} \mathbb{E}\left[\mathcal{E}\left(V_\phi(s_t),\ Z_t^{(\lambda)}\right)\right]$$

    via stochastic gradient descent.

**end**

---

The clipping function $g(\epsilon, A)$ shown in the algorithm is defined as follows:

$$g(\epsilon, A) = \begin{cases} (1+\epsilon)A & \text{if } A \geq 0 \\ (1-\epsilon)A & \text{if } A < 0 \end{cases}$$

Note that expectation of each loss is taken over each the collection of trajectories and environments.

# F    IMPLEMENTATION DETAILS

For producing a categorical distribution, a softmax layer was added to the output of the network. For producing a Gaussian mixture distribution, the mean of each Gaussian is simply the output of the network, the variance is kept positive by running the output through a softplus layer, and the weights of each Gaussian is produced through the softmax layer. Since our proposed method takes an archi-

Table 2: Network architecture for GMAC on atari

| Layer Type | Specifications | | Filter size, stride |
|---|---|---|---|
| Input | 84 x 84 x 4 | | |
| Conv1 | 20 x 20 x 32 | | 8 x 8 x 32, 4 |
| Conv2 | 9 x 9 x 64 | | 4 x 4 x 64, 2 |
| Conv3 | 7 x 7 x 32 | | 3 x 3 x 32, 1 |
| FC1 | 512 | | |
| Heads | Policy | Value | |
| (FC) | action dim | # of modes $(= 5)$ | |

tecture which only changes the value head of the original PPO network, we base our hyperparameter settings from the original paper (Schulman et al., 2017). We performed a hyperparameter search on a subset of variables: optimizers={Adam, RMSprop}, learning rate={2.5e-4, 1.0e-4}, number of epochs={4, 10}, batch size={256, 512}, and number of environments={16, 32, 64, 128} over 3 atari tasks of Breakout, Gravitar, and Seaquest, for which there was no degrade in the performance of PPO.

Table 3: Parameter settings for training atari games

| Parameter | PPO | IQAC | IQAC-E | GMAC |
|---|---|---|---|---|
| Learning rate | | | 2.5e-4 | |
| Optimizer | | | Adam | |
| Total frames | | | 2e8 | |
| Rollout steps | | | 128 | |
| Skip frame | | | 4 | |
| Environments | | | 64 | |
| Minibatch size | | | 512 | |
| Epoch | | | 4 | |
| $\gamma$ | | | 0.99 | |
| $\lambda$ | | | 0.95 | |
| Dirac samples | - | | 64 | - |
| Mixtures | - | - | - | 5 |

Table 4: Parameter settings for training PyBullet games

| Parameter | PPO | IQAC | IQAC-E | GMAC |
|---|---|---|---|---|
| Learning rate | | | 1e-4 | |
| Optimizer | | | Adam | |
| Total env steps | | | 5e-7 | |
| Rollout steps | | | 512 | |
| Skip fram | | | 1 | |
| Environments | | | 64 | |
| Minibatch size | | | 2048 | |
| Epoch | | | 10 | |
| $\gamma$ | | | 0.99 | |
| $\lambda$ | | | 0.95 | |
| Dirac samples | - | | 64 | - |
| Mixtures | - | - | - | 5 |

# G   MORE EXPERIMENTAL RESULTS

## G.1   FIVE-STATE MDP

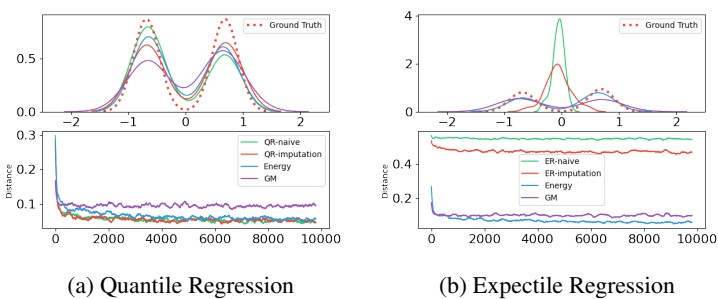

(a) Quantile Regression        (b) Expectile Regression

Figure 4: In addition to 2, quantile and expectile regressions are also evaluated in the 5-state MDP with imputation using neural networks architectures of implicit quantile network against IQE and Gaussian mixture with their respective loss functions.

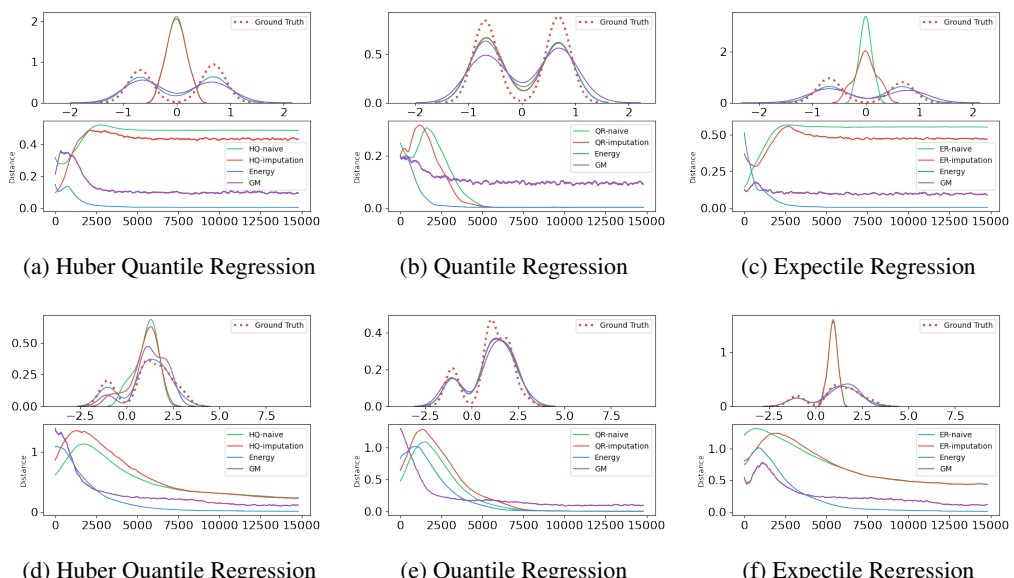

(a) Huber Quantile Regression     (b) Quantile Regression     (c) Expectile Regression

(d) Huber Quantile Regression     (e) Quantile Regression     (f) Expectile Regression

Figure 5: Evaluation of the five-state MDP under tabular setting on a symmetric reward distribution (top row) and on assymetric reward distribution (bottom row). Huber quantile($\kappa = 1$), quantile, and expectile regressions are compared to the energy distance minimization between samples and Gaussian mixtures.

Here we provide more details on the five-state MDP presented in Figure 2. For each cases in the figure, 15 diracs are used for quantile based methods and 5 mixtures are used for GMM to balance the total number of parameters required to represent a distribution. For the cases with the label "naïve", the network outputs (quantiles, expectiles, etc.) are used to create the plot. On the other hand, the cases with "imputation" labels apply appropriate imputation strategy to the statistics to produce samples which are then used to plot the distribution. In all cases, the network is trained on full batches of all states for 8 different empirical samples with Adam optimizer and learning rate of 1e-3. Sample based energy-distance was used to calculate the distance from the true distribution for all cases. The tabular setting of five-state MDP in Figure 5 uses same exact experimental set-up as the one with neural network. The only difference is that the statistics are from evenly spaced quantile/expectile targets $\tau \in [0, 1]$.

## G.2 Value distribution learning in Atari games

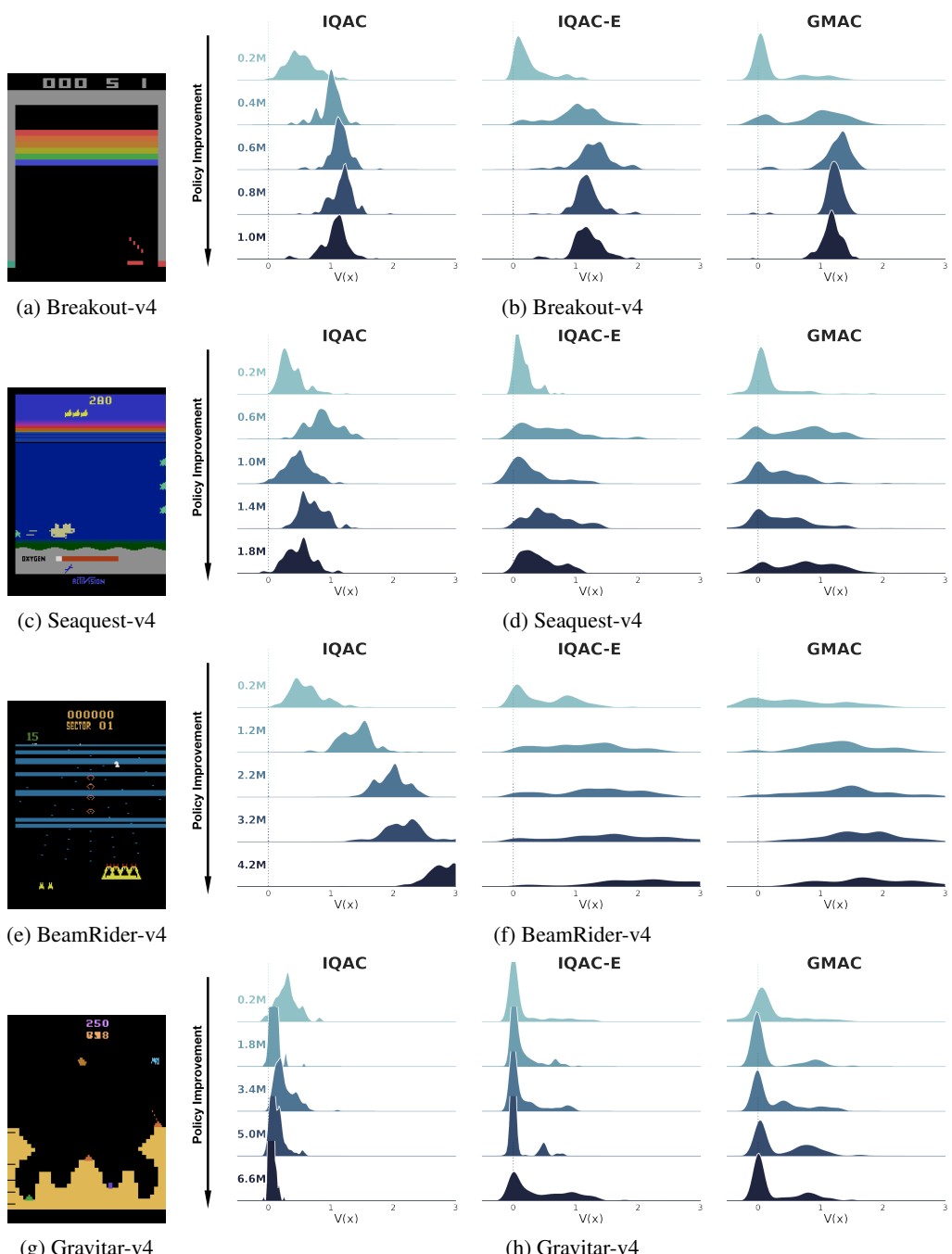

Figure 6: More value distributions of different tasks. All states are chosen such that the agent is in place of near-death or near positive score. Thus, when the policy is not fully trained, such as in a very early stage, the value distribution should include a notion of death indicated by a mode positioned at zero. In all games, IQN + Huber quantile (IQAC) fails to correctly capture a mode positioned at zero while the other two methods, IQN + energy distance (IQAC-E) and GMM + energy distance (GMAC) captures the mode in the early stage of policy improvement. Again, the visual representation is *maxpool* of the 4 frame stacks in the given state.

## G.3 LEARNING CURVES

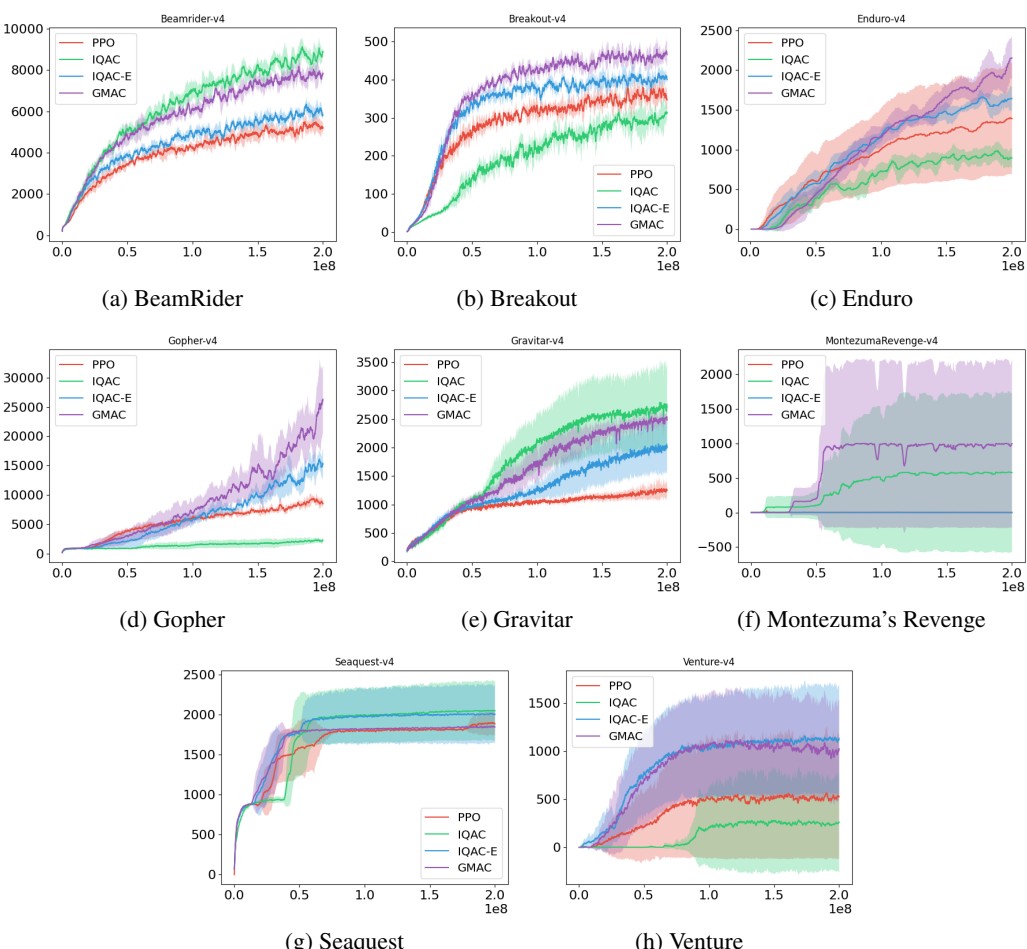

Figure 7: Raw learning curves for 8 selected atari games.

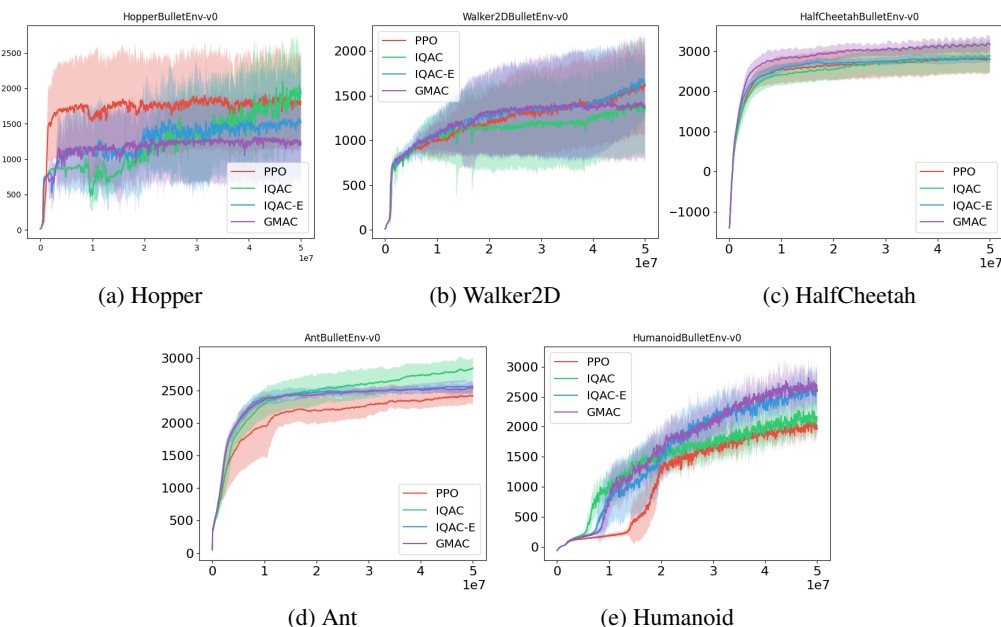

Figure 8: Raw learning curves for 5 selected PyBullet continuous control tasks.

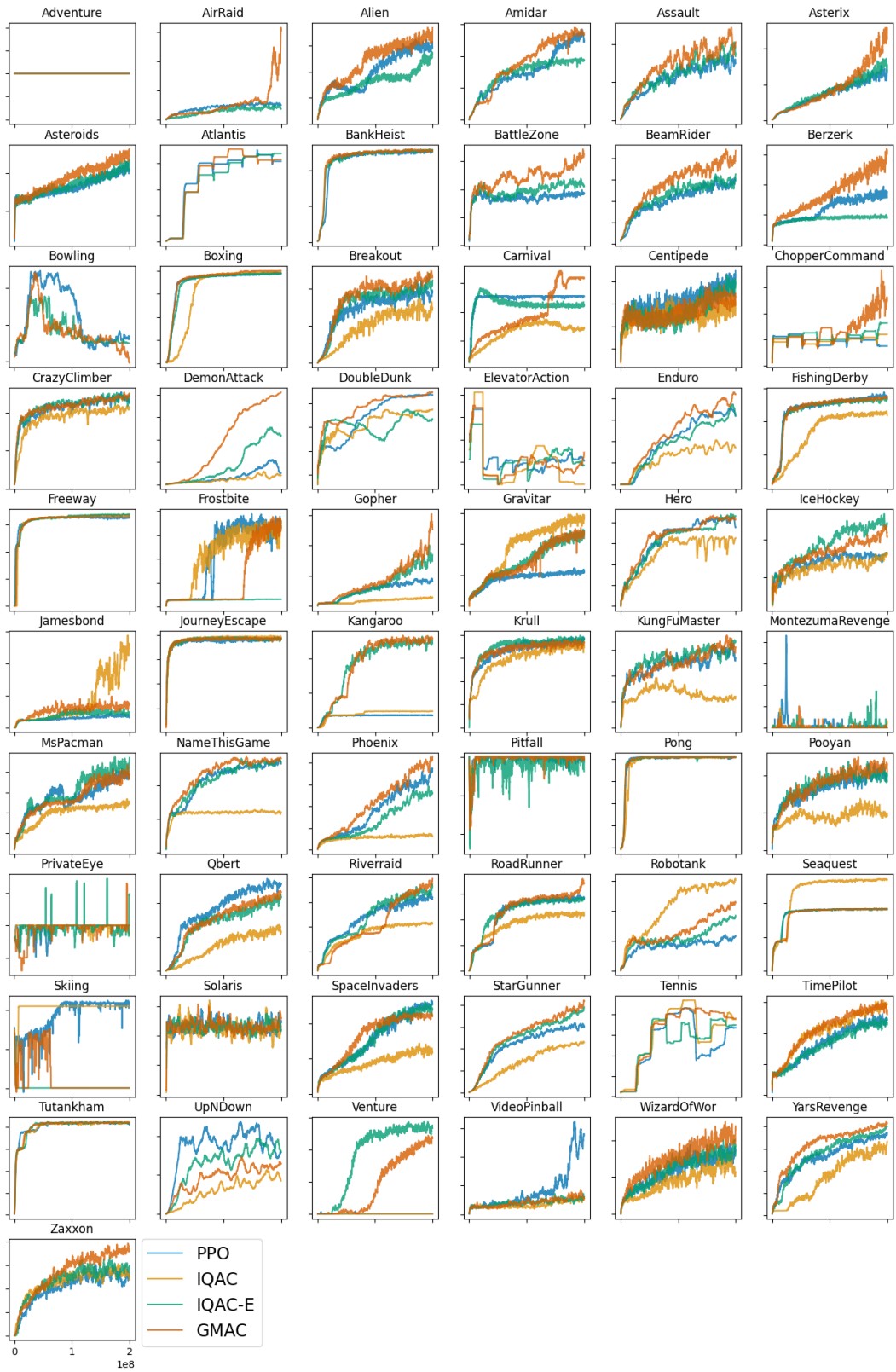

Figure 9: Full learning curves of 61 atari games from ALE

Table 5: Average score over last 100 episodes in 200M frame collected for training 61 atari games. The algorithms are trained using same single seed and hyperparameters. Random and Human scores are taken from Wang et al.

| GAMES | RANDOM | HUMAN | PPO | IQAC-E | GMAC |
|---|---|---|---|---|---|
| Adventure | NA | NA | **0.00** | **0.00** | **0.00** |
| AirRaid | NA | NA | 10,205.75 | 7,589.50 | **62,328.75** |
| Alien | 227.80 | **7,127.70** | 2,918.60 | 2,704.20 | 3,687.10 |
| Amidar | 5.80 | **1,719.50** | 1,244.12 | 932.11 | 1,363.72 |
| Assault | 222.40 | 742.00 | 7,508.03 | 8,589.55 | **10,281.73** |
| Asterix | 210.00 | 8,503.30 | 13,367.00 | 15,426.00 | **22,650.00** |
| Asteroids | 719.10 | **47,388.70** | 2,088.10 | 2,332.00 | 2,597.50 |
| Atlantis | 12,850.00 | 29,028.10 | 3,073,796.00 | **3,373,635.00** | 3,141,534.00 |
| BankHeist | 14.20 | 753.10 | 1,263.80 | **1,286.60** | 1,274.30 |
| BattleZone | 2,360.00 | **37,187.50** | 18,540.00 | 21,310.00 | 32,490.00 |
| BeamRider | 363.90 | **16,926.50** | 5,913.84 | 6,507.68 | 8,718.72 |
| Berzerk | 123.70 | 2,630.40 | 1,748.10 | 887.50 | **3,081.20** |
| Bowling | 23.10 | **160.70** | 33.54 | 30.00 | 19.39 |
| Boxing | 0.10 | 12.10 | 96.79 | 97.10 | **99.89** |
| Breakout | 1.70 | 30.50 | 384.29 | 445.64 | **462.68** |
| Carnival | NA | NA | 5,079.20 | 4,401.00 | **6,344.20** |
| Centipede | 2,090.90 | **12,017.00** | 5,205.25 | 4,864.69 | 4,303.10 |
| ChopperCommand | 811.00 | **7,387.90** | 872.00 | 1,314.00 | 1,795.00 |
| CrazyClimber | 10,780.50 | 35,829.40 | 112,640.00 | 121,550.00 | **125,143.00** |
| DemonAttack | 152.10 | 1,971.00 | 50,590.65 | 236,839.85 | **411,118.85** |
| DoubleDunk | -18.60 | -16.40 | -3.26 | -8.28 | **-2.72** |
| ElevatorAction | NA | NA | 10,449.00 | 8,516.00 | **14,254.00** |
| Enduro | 0.00 | 860.50 | 1,588.68 | 1,612.17 | **2,092.65** |
| FishingDerby | -91.70 | -38.70 | 37.01 | 33.13 | **37.52** |
| Freeway | 0.00 | 29.60 | 32.53 | **33.68** | 32.84 |
| Frostbite | 62.50 | **4,334.70** | 3,571.50 | 307.10 | 3,392.40 |
| Gopher | 257.60 | 2,412.50 | 8,199.80 | 16,934.60 | **25,266.80** |
| Gravitar | 173.00 | **3,351.40** | 1,151.50 | 2,178.50 | 2,401.00 |
| Hero | 1,027.00 | 30,826.40 | 37,725.55 | **43,065.95** | 41,509.05 |
| IceHockey | -11.20 | 0.90 | -1.90 | **2.13** | 0.34 |
| Jamesbond | 29.00 | 302.80 | 642.50 | 961.00 | **1,512.00** |
| JourneyEscape | NA | NA | **-607.00** | -840.00 | -680.00 |
| Kangaroo | 52.00 | 3,035.00 | 1,742.00 | 12,208.00 | **12,909.00** |
| Krull | 1,598.00 | 2,665.50 | **9,605.51** | 9,514.03 | 9,127.63 |
| KungFuMaster | 258.50 | 22,736.50 | 26,846.00 | 33,378.00 | **31,025.00** |
| MontezumaRevenge | 0.00 | **4,753.30** | 0.00 | 0.00 | 0.00 |
| MsPacman | 307.30 | **6,951.60** | 3,674.20 | 4,699.00 | 3,884.40 |
| NameThisGame | 2,292.30 | 8,049.00 | 13,229.10 | 13,454.00 | **14,031.30** |
| Phoenix | 761.40 | 7,242.60 | 37,263.70 | 26,154.00 | **42,664.00** |
| Pitfall | -229.40 | **6,463.70** | 0.00 | -18.86 | -3.36 |
| Pong | -20.70 | 14.60 | 20.87 | 20.88 | **20.97** |
| Pooyan | NA | NA | 4,018.95 | 3,674.70 | **4,178.65** |
| PrivateEye | 24.90 | **69,571.30** | 100.00 | 196.30 | 100.00 |
| Qbert | 163.90 | 13,455.00 | **25,519.25** | 21,599.50 | 23,176.25 |
| Riverraid | 1,338.50 | 17,118.00 | 15,983.00 | 18,073.40 | **19,761.30** |
| RoadRunner | 11.50 | 7,845.00 | 56,321.00 | 56,121.00 | **68,272.00** |
| Robotank | 2.20 | 11.90 | 23.45 | 36.69 | **45.82** |
| Seaquest | 68.40 | **42,054.70** | 1,832.00 | 1,814.60 | 1,838.40 |
| Skiing | -17,098.10 | **-4,336.90** | -7,958.81 | -29,971.02 | -29,975.52 |
| Solaris | 1,236.30 | **12,326.70** | 2,452.80 | 2,204.80 | 2,579.20 |
| SpaceInvaders | 148.00 | 1,668.70 | **2,544.10** | 2,410.90 | 2,228.30 |
| StarGunner | 664.00 | 10,250.00 | 74,848.00 | 97,450.00 | **104,188.00** |
| Tennis | -23.80 | -8.30 | -8.16 | -7.54 | **-5.90** |
| TimePilot | 3,568.00 | 5,229.20 | 12,157.00 | 11,704.00 | **13,227.00** |
| Tutankham | 11,40 | 167.60 | 206.32 | 208.72 | **209.82** |
| UpNDown | 533.40 | 11,693.20 | 158,629.50 | **161,328.40** | 129,243.70 |
| Venture | 0.00 | 1,187.50 | 0.00 | **1,339.00** | 1,181.00 |
| VideoPinball | 16,256.90 | 17,667.90 | **279,504.81** | 59,988.90 | 55,272.82 |
| WizardOfWar | 563.50 | 4,756.50 | 8,749.00 | 9,165.00 | **11,388.00** |
| YarsRevenge | 3,092.90 | 54,576.90 | 92,709.94 | 100,082.55 | **103,895.05** |
| Zaxxon | 32.50 | 9,173.00 | 13,336.00 | 14,882.00 | **18,436.00** |

