# OpenReview forum: "A Distributional Perspective on Actor-Critic Framework"
_ICLR.cc/2021/Conference — Reject_

### Official Review · AnonReviewer2 · 2020-10-27
**Interesting application of distributional RL to actor-critic, although with scope for further empirical comparison**

**Rating:** 5
**Confidence:** 5

**Review:**

Summary

This paper proposes an actor-critic algorithm based on distributional RL. Distributions are approximated as mixtures of Gaussians, Bellman targets are computed using lambda-returns, and the training loss is computed using an energy distance.

Review Summary

There is a solid algorithmic contribution in this paper, although at present the comparison against baselines considered feels a little incomplete. I would be happy to review my rating if further results are reported.

Significance

While several aspects of the proposed algorithm have been explored before (multi-step returns in distributional RL, Cramer distance in training loss), these components are combined to yield a novel algorithm, and the treatment of multi-step returns differs from previous work.

Quality & Clarity

The proposed method is presented reasonably clearly in Algorithm 1 and related discussion in Section 4. There is some looseness of mathematical notation in Section 4, such as the use of densities, and the discussion around Equation (18), but broadly the paper is clear.

Technical comments

The paper mentions three issues with DRL: instability in the control setting, specificity of algorithms to certain action spaces, and biasedness in sample gradients. I think the first two points stand, but am less sure about the third. Wasserstein distances have been used in the analysis of distributional RL algorithms, but as far as I know in general they are not used algorithmically. Many distributional algorithms use methods that do yield unbiased gradients, such as the quantile regression approach used by QR-DQN, IQN, and FQF, and the Cramer distance used by S51 in "Distribution Reinforcement Learning with Linear Function Approximation". This issue is mentioned again below Equation (14) in comparison with IQN, which uses a quantile regression loss, rather than an empirical Wasserstein distance loss. Can the authors comment on this?

Discussion around Equation (18): Is this a formal claim (that the density of Z(x, a) can be approximated by a Gaussian mixture distribution)? It seems as though without being quantitative, this argument could be made for any distribution with a density. As an additional note, what if Z(x, a) does not have a density?

Experiment hyperparameters: How were the hyperparameters (e.g. learning rate, Adam epsilon, lambda, etc.) selected for the methods considered in the paper? Were any hyperparameter sweeps undertaken? What does "Epoch" in Tables 3 and 4 refer to?

Figure 2 experiment. The text mentions that minimizing the Wasserstein distance (labelled as IQ) leads to a local minimum. Can the authors clarify whether they are minimizing the Wasserstein distance between approximate distributions, or minimizing a quantile regression loss?

The intrinsic motivation experiments are interesting. I couldn't find details in the appendix as to precisely how the intrinsic reward is used - is the raw TD/Cramer distance used as a reward/is there any scaling applied? Is the Cramer error typically of a different magnitude to the TD error? One might expect the raw TD error to be of a very low magnitude, which may mean that it requires scaling before use as an intrinsic reward (although of course scaling introduces an additional hyperparameter).

How were the Atari results displayed in Figures 3 & 6 selected? It looks as though in all games in Figure 3, GMAC comes out on top, although judging from Figure 6 this is not always the case. Can the authors include the results of IQAC/IQAC-E in Table 5? Can the authors report summary statistics for the performance of the algorithms across the suite of games considered, such as human-normalized mean/median performance? At present the comparison of GMAC against the baselines feels a little incomplete; as far as I can tell, full results for IQAC/IQAC-E are not reported in the paper.

GMAC implementation details. I couldn't find precise details on the architecture used for GMAC, in particular the outputs of the network. Presumably each mixture component requires 3 heads (for mixture weight, mean, and variance/stddev). Is a softmax used to ensure the sum of the weights is 1? How is non-negativity of the variance parameters enforced?

IQN-based agent implementation details. Is the embedding of tau as in Equation (4) of "Implicit Quantile Networks for Distributional Reinforcement Learning", but with n=32?

Minor comments

"Note that the cdf of \tilde{Z} has a different domain from \tilde{F}_Z". This wasn't clear to me, can the authors expand?

Below Equation (12), the authors discuss pdfs of Z^{(n)}_t etc., but presumably these random variables may not have pdfs?

I didn't understand the comment that the cdf F_{Z^{(\lambda)}_t} has a simple form, and that evaluating requires O(n^2) time and memory. Presumably if the CDFs that appear in the mixture in Equation (12) don't have a simple form, then neither will F_{Z^{(\lambda)}_t}? Can the authors give more detail on where O(n^2) time and space complexity come from?

Section 4.3: "The reward given a state-action pair R(x, a) follows a single distribution with finite variance". What is meant by "single distribution" here?

Equation (19): I think there are missing factors of 1/2 in front of the \delta(X, X') and \delta(Y, Y') terms.

Minor comments on formatting etc.
Consider using \eqref rather than \ref when referring to equations.
References should point to conference versions of papers (rather than arxiv) where appropriate (e.g. WGAN reference).



Post-rebuttal update

Overall, I am still borderline on this paper. I appreciate the effort the authors have put in during the rebuttal phase, and would say the paper is now clearer.
The inclusion of mean/median normalized scores for the Atari results in the main paper has improved the experimental section. However, the main drawbacks that remain are empirical; the smaller scale comparisons between methods need to be updated to give a like-for-like comparison in terms of numbers of parameters etc., as the authors acknowledge, and the paper still lacks baseline distributional agents in the large-scale experiments. This is important, as it means it is difficult to assess the impact that, for example, SR(lambda) may be having on the experimental results.

---

> ### Author Response · Authors · 2020-11-19
> **Comments for AnonReviewer2**
>
> Thank you for the thorough review of our paper. Here are our comments about your questions.
>
> > The paper mentions three issues with DRL:...Can the authors comment on this?
>
> As pointed out in the question, the Huber quantile function which we used to compare our method to in the text is not the same as the Wasserstein distance. Thus it was wrong and misleading to have treated the Huber quantile loss as the Wasserstein distance. Therefore, we correct our statement on the biased gradients to stating the data conflation as the main cause of the problem of the regression performance of the Huber quantile, for which the details can be found in Rowland et al. (2019).
> To be more clear about this argument, we have modified the toy task of the two-state MDP problem to five-state MDP in which the conflation of statistics and samples is more apparent. When extending the length of the MDP chain, the underestimation of variance from the conflation becomes clearer as seen in the graph of Huber quantile (Fig. 2(b)). However, this phenomenon is not present when minimizing energy distance. More results added in appendix F also paint a similar picture, and one can see that the quantile regression (L1 loss) does not suffer from the conflation.
>
> > Discussion around Equation (18): Is this a formal claim...what if Z(x, a) does not have a density?
>
> Thank you for pointing out this subtlety. After revising our arguments on the distributional analogy in Section 4.1, we came to realize that our approximation of probability measures does not require a density function; the revised version now appeals to the existence of dense sets in the set of finite signed Borel measures equipped with the weak* topology (Bogachev, 2007). Under this statement, both the point-masses and convex combinations of Gaussian measures can be used to approximate Z(x,a), without the assumption of its density function.
>
> > Experiment hyperparameters:...What does "Epoch" in Tables 3 and 4 refer to?
>
> The details on the hyperparameter search have been added to the appendix E. The term “epoch” used here refers to the number of epochs to sweep through the entire batch collected for the n-step rollouts across all training environments.
>
> > Figure 2 experiment. The text mentions that minimizing the Wasserstein...or minimizing a quantile regression loss?
>
> This relates back to the answer to the first question. The loss is the (Huber) quantile loss and not the Wasserstein distance. To correct the argument with Wasserstein distance, as discussed in the answer to the first question, we change the toy task of Figure 2. to a 5-state MDP, to display more clearly the effect of conflation from successive Bellman operations. The newly updated experimental results show that the minimization of Huber quantile loss leads to a false distribution while our method does not suffer from such issues since they use “samples” or “parameters of the distribution” instead of statistics, as the parameters being minimized through the loss functions.
>
> > The intrinsic motivation experiments...(although of course scaling introduces an additional hyperparameter).
>
> For the intrinsic motivation experiment section, both the TD error and Cramer distances were normalized through the RewardForwardFilter which was used in Burda et. al., 2018. Therefore two metrics are of equal scale.
>
> > How were the Atari results displayed in Figures 3 & 6 selected?...not reported in the paper.
>
> As for the selection in Figures 3 and 6, the tasks were chosen on two folds: ones that have shown increased performance between expectation learning algorithms (e.g. DQN) and distribution learning algorithms (e.g. IQN), and ones in which the task itself has some stochasticity. Due to the concerns on the fairness of the selection, we have included the full learning curve of 61 atari games for PPO, IQACE, and GMAC. IQAC was not considered for the full run because throughout other experiments, applying Huber quantile without additional imputation leads to a false representation of the distribution and thus we could not form any well-founded interpretations about its performance. Additionally, human normalized learning curves are also included in the main text. Results from additional seeds, including the results from IQAC, will be added as soon as they are prepared.
>
> > GMAC implementation details...How is non-negativity of the variance parameters enforced?
>
> For the details on the network architecture of GMAC, a softmax function was used to enforce the sum of weight to equal 1. As for the variance, a softplus function was used to ensure that the predicted variances are always positive.
>
> > IQN-based agent implementation details. Is the embedding of tau as in Equation (4) of "Implicit Quantile Networks for Distributional Reinforcement Learning", but with n=32?
>
> The IQN implementation followed the implementation details in the IQN paper, which chose n=64.

---

> > ### Author Response · Authors · 2020-11-19
> > **Comments for AnonReviewer2 - continued**
> >
> > > "Note that the cdf of \tilde{Z} has a different domain from \tilde{F}_Z". This wasn't clear to me, can the authors expand?
> >
> > This statement in the paper was erroneous and does not add additional information towards our explanation, so we have taken this out while rephrasing some statements in Section 4.
> >
> > > Below Equation (12), the authors discuss pdfs of Z^{(n)}_t etc., but presumably these random variables may not have pdfs?
> >
> > Similar to the answer to the second question, we no longer require the random variables to have pdfs to develop our arguments. Accordingly, we have also modified the development of our argument to show that the expectation of the distributional lambda-return is equal to the scalar lambda return.
> >
> > > I didn't understand the comment that the cdf F_{Z^{(\lambda)}t}...where O(n^2) time and space complexity come from?
> >
> > It was misleading to say that the time and space complexities are O(n^2) in practice. Our intention was that calculating Z^{(\lambda)}_t requires an order of n^2 different distributions, i.e., n different sets of Z^{(n)}_t for t=T:T+N. Calculating them will also require looping through in the order of n^2. However, as soon as we reformulize the process with empirical cdfs with finite samples, the memory complexity becomes O(n) as each distribution can be formulated as a linear combination of the N different samples. SR(lambda) takes this further to ensure the time complexity becomes O(n) as well. We have elaborated on this expression also in the main text.
> >
> > > Section 4.3: "The reward given a state-action pair R(x, a) follows a single distribution with finite variance". What is meant by "single distribution" here?
> >
> > This was also very unclear and was taken out of the context, along with the revision to the arguments about the pdfs and densities of random variables.
> >
> > Revision to the issues mentioned above, along with other minor comments, have been made in the main text.
> >
> > > Consider using \eqref rather than \ref when referring to equations.
> >
> > The reference for equations in the main text already uses \eqref. Could you please elaborate on this statement?
> >
> > > References should point to conference versions of papers (rather than arxiv) where appropriate (e.g. WGAN reference).
> >
> > Thank you for pointing out. We have changed the versions accordingly.
> >
> >
> > **References**
> >
> > [1] Mark Rowland, Robert Dadashi, Saurabh Kumar, R ́emi Munos, Marc G. Bellemare, and Will Dab-ney. Statistics and samples in distributional reinforcement learning. ICML, 2019.
> >
> > [2] Vladimir I Bogachev. Measure theory, volume 1. Springer Science & Business Media, 2007.

---

> > > ### Comment · AnonReviewer2 · 2020-11-23
> > > **Response to authors**
> > >
> > > I thank the authors for their extensive comments, and the revisions of the paper. I'm glad that certain parts of the main text have been clarified and further experimental details have been given.
> > >
> > > There are still some parts of the paper that I think need to be updated in line with the clarifications the authors have given in their response. For example, there are still some parts of the paper that refer to the bias associated with the Wasserstein distance, such as the abstract and the third paragraph of the introduction, which could be updated. Section 4.1 seems to lapse into using pdf notation at around Equation (13).
> > >
> > > In response to the \eqref question, there are terms like:
> > > "Instead, we redefine equation 10 in terms of cdfs:"
> > > in the paper, where it looks as though \ref has been used rather than \eqref.
> > >
> > > I also have a few questions about some of the new material added to the paper:
> > >
> > > I found the new discussion around weak* topology unclear. It seems as though two unrelated points are being made: one, that the particular sets of distributions are dense in the space of Borel probability measures over R under the weak* topology, and then a separate point that an algorithmic choice is made to represent distributions with a finite, fixed number of Diracs. At present, this paragraph reads as though the denseness of a certain set of distributions under weak* topology serves as a formal justification for the approximation made in the algorithm. Can the authors clarify whether these two points are formally connected?
> > >
> > > I also have a few questions about the new 5-state MDP experiments:
> > > Figure 2a: I think the reward description in this picture doesn't match up with the ground truth density plotted in Figure 2a; the authors may mean N(0, 0.1^2) for the final reward distribution?
> > > Figure 2b:
> > > Can the authors describe which parameter choices were used (number of particles/mixture components etc.) for each the methods compared?
> > > What is the distance measured on the y-axis? Is it Cramer/energy distance? Is this the same for the experiments in Appendix G.1?
> > > In the case of IQ-imputation, are you plotting the density derived from treating the Huber quantiles as samples, or from the imputed samples themselves?
> > > Can the authors describe the difference between QR and QR-Naive in Figure 5b and 5e?

---

> > > > ### Author Response · Authors · 2020-11-24
> > > > **Reply to the second response from AnonReviewer2**
> > > >
> > > > Thank you for the reply. We are glad as well that our feedback has resolved some of the unclarities that you had.
> > > >
> > > > > there are still some parts of the paper that refer to the bias associated with the Wasserstein distance, such as the abstract and the third paragraph of the introduction, which could be updated.
> > > >
> > > > We have made the appropriate changes. Thank you.
> > > >
> > > > > Section 4.1 seems to lapse into using pdf notation at around Equation (13).
> > > >
> > > > We have made changes to section 4.1 along with the statements about weak* topology.
> > > >
> > > > > In response to the \eqref question, there are terms like: "Instead, we redefine equation 10 in terms of cdfs:" in the paper, where it looks as though \ref has been used rather than \eqref.
> > > >
> > > > It seems that the mathcommnad file provided with the ICLR2021 style format has set the /eqref to display equation references in form of “equation #” instead of “(#)”, and we think this is the source of confusion. Currently, we are sticking with the provided format but if this makes the text difficult to read, we can change it.
> > > >
> > > > > I found the new discussion around weak* topology unclear...Can the authors clarify whether these two points are formally connected?
> > > >
> > > > We agree with what you pointed out. Therefore, we have removed our attempt to give justification for using a mixture of finite samples as the approximation. We find this misleading as you have pointed out and not giving any justifications for using only a finite number of samples. Instead, we build our algorithm upon the assumption where we approximate a distribution using different parametric models, which have been used in previous literatures. The end of section 4.1 and beginning of section 4.3 have been modified to reflect this.
> > > >
> > > > > Figure 2a: I think the reward description in this picture doesn't match up with the ground truth density plotted in Figure 2a; the authors may mean N(0, 0.1^2) for the final reward distribution?
> > > >
> > > > Yes this was a typo. Thank you for spotting it.
> > > >
> > > > > Figure 2b: Can the authors describe which parameter choices were used (number of particles/mixture components etc.) for each the methods compared? What is the distance measured on the y-axis? Is it Cramer/energy distance? Is this the same for the experiments in Appendix G.1?
> > > >
> > > > Details of parameter choices have been added in the appendix. Just to list some here, the distance is the energy distance and 8 parameters/mixtures were used to approximate each distribution for all settings and methods.
> > > >
> > > > > In the case of IQ-imputation, are you plotting the density derived from treating the Huber quantiles as samples, or from the imputed samples themselves?
> > > >
> > > > The ones labeled “naive” plot Huber quantiles, while the ones labeled “imputation” are the imputed samples.
> > > >
> > > > > Can the authors describe the difference between QR and QR-Naive in Figure 5b and 5e?
> > > >
> > > > Just to make sure, do you mean the difference between QR-naive and QR-imputation?
> > > > QR-Naive refers to the QR without any imputation strategy. On the other hand, QR-imputation uses the imputation strategy described in Rowland et al., 2019, which should not make much difference and does not indeed show any significant change compared to the naive setting of just QR.

---

> > > > > ### Comment · AnonReviewer2 · 2020-11-24
> > > > > **Response to authors**
> > > > >
> > > > > Thanks for the clarifications. Responses to a few particular points below:
> > > > >
> > > > > Thanks for clarifying re \eqref - I would be happy with whichever choice the authors make here.
> > > > >
> > > > > On the experiments in Figure 2, if I understand correctly, it sounds as though the Gaussian mixture distributions in these experiments use approximately 3 times more parameters than the quantile experiments (since each mixture has a weight, location, and scale parameter) - is this correct? If so, a fairer comparison might be to use a similar number of quantiles as parameters in the mixture model. It would also be interesting to see the tabular experiments run to convergence, since from the energy distance plots, it looks as though several of the methods are still making improvements at the end of the updates.
> > > > >
> > > > > I will update my original review once the reviewer discussion has finished.

---

> > > > > > ### Author Response · Authors · 2020-11-24
> > > > > > **On the experiments for the five-state MDP**
> > > > > >
> > > > > > > On the experiments in Figure 2, ... - is this correct?
> > > > > >
> > > > > > Yes, the GMAC uses 3 times more parameters for the experiments shown in Fig. 2, 4, and 5. Although the total number of parameters for the whole network is in fact smaller in GMAC than the quantile models in Fig. 2 and 4, we agree with your point that to give a fair comparison of the ability for correct representation, the number of parameters should match across the methods (especially for the tabular case shown in Fig. 5). We shall run the above experiments until convergence, and update the three figures along with the results from IQAC run on atari games possibly before the discussion phase ends, otherwise, they will be included for future revisions.
> > > > > >
> > > > > > We thank you for your time and constructive feedback throughout the process.

---

### Official Review · AnonReviewer1 · 2020-10-29
**An ambitious AC framework trying to solve 3 problems with Distributional RL**

**Rating:** 7
**Confidence:** 5

**Review:**

This paper proposed a Gaussian-mixture Actor Critic (GMAC) framework to address the problems of distributional RL (distRL): distributional instability, action-type restrictions, and biased approximation (replacing Wasserstein distance with Cramer distance. The framework uses Gaussian mixtures to represent the distribution of value functions, and learn the Gaussian distribution parameters.

Using GMMs for distRL: it appears explored before in ref 1, but not cited.

my problem of GMMs: the model may be not Gaussian. The distribution learned by DistRL is prevoulsy shown to be asymmetirc (Fig 5 of Mavrin et. al. 2019).

one question: since the three problems also exist for distRL alg's like QRDQN, why don't you improve the original value function approximation distRL alg's, but instead on AC algorithms?

DRL:using it for distributional rl is a bit unconventional. DRL: deep reinforcement learning.


Morimura et al. (2010a;b) designed a risk-sensitive algorithm using a distributional perspective: this paper is perhaps the earliest concept of distRL.


Mavrin et al. (2019) utilized the idea of the uncertainty captured from the variance of value distribution with a decay factor to add an intrinsic reward to the objective of conventional greedy policy: "utilized the idea of the uncertainty captured from the variance of value distribution" is correct. DLTV uses the distribution/quantiles/variance to estimate the upper confidence bound and use it for action selection. It's not "adding an intrinsic reward".

our work is the first to connect stochastic policy as a solution to
the problems in value-based DRL: this isn't very clear.

We believe that the findings from this paper can easily generalize
to other actor-critic frameworks as well: are you sure the other alg's all have the same kind of the (three) issues?

Below eq 15:
do you mean Bellman optimality operator is a contraction mapping? It appears so from Appendix C. this is interesting because with Wasserstein distance the Bellman optimality operator is not a contraction in any norm (pls confirm this is correct).

Experiments were performed on two-state MDP (to test the Cramer loss func is less biased while Huber loss function used in QRDQN is biased),

Representing Multimodality section:
Here the experiment lacks a study of assymetric models where GMMs cannot represents. Previously experiments by Mavrin et. al. 2019 showed in Pong, e.g., the Q value function distribution is not symmetric.

Discrete and Continuous Action Spaces section:
Do you use the same algorithm for both cases?	Why PPO, IQAC IQAC-E GMAC are selected? How these alg's compare to distRL alg's like DQN, C51, QRDQN?


Ref 1:
Distributional Deep Reinforcement Learning
with a Mixture of Gaussians
http://cpslab.snu.ac.kr/publications/papers/2019_icra_ddrl_mog.pdf

---

> ### Author Response · Authors · 2020-11-19
> **Comments for AnonReviewer1**
>
> Thank you for the constructive review and detailed comments.
>
> While we aim to answer all of the questions that you have raised, we would like to start off with a comment regarding the Wasserstein distance and the Huber quantile loss. As mentioned in the reviews, solving an optimization problem using the Huber quantile function is not the same as using the Wasserstein distance. It was wrong, and misleading to treat minimizing the Huber quantile loss as if it is equal to minimizing the Wasserstein distance. Therefore, we have corrected our statement about the cause of biased representations, as the conflation between statistics and samples when using the Huber quantile loss.
>
> > Using GMMs for distRL: it appears explored before in ref 1, but not cited.
>
> The attribution to MoG-DQN (Choi, 2019) has been added in Section 2 (Related Works) of the main text. Thank you for mentioning closely related literature.
>
> > the model may not be Gaussian. The distribution learned by DistRL is previously shown to be asymmetric (Fig 5 of Mavrin et. al. 2019).
>
> The convex combination of Gaussian mixtures is known to be dense in the set of finite signed Borel measures in the real line, as now stated above eq (13). This assumption of measures includes square-integrable density functions, which include the asymmetrical function (exponential density function) shown in Fig 5 of Mavrin et. al. 2019.
>
> > one question: since the three problems also exist for distRL alg's like QRDQN, why don't you improve the original value function approximation distRL alg's, but instead on AC algorithms?
>
> Of the three problems mentioned, two of them really apply to the value-based algorithms like QR-DQN, IQN, etc. which are the instability issue and limitation in action spaces. The instability issue is not present in the evaluation setting of the Bellman operator. Therefore, an easy way to eliminate the instability is to introduce a parameterized policy and use the evaluation setting only. This thought naturally led us to consider the problem formulation within the domains of actor-critic methods. Furthermore, in general, actor-critic methods are not constrained by the action space types as well.
> Whereas the value-based methods, like the QR-DQN, handling infinitely many actions is infeasible and requires changes in architecture, thus was not considered as the baseline algorithm to build our method upon.
>
> > DRL: using it for distributional rl is a bit unconventional. DRL: deep reinforcement learning.
>
> For the convention of DRL, we do acknowledge that in general RL, the acronym DRL normally stands for deep reinforcement learning. However, following the previous works that play a major part in formulating the arguments of our paper, we followed their convention of using DRL to represent distributional reinforcement learning instead. If this may cause a problem, we will change to follow the usual convention.
>
> > Morimura et al. (2010a;b) designed a risk-sensitive algorithm using a distributional perspective: this paper is perhaps the earliest concept of distRL.
>
> Thank you for pointing out. The comment has been added in Section 2 (Related Works).
>
> > Mavrin et al. (2019) utilized the idea of the uncertainty captured from the variance of value distribution with a decay factor to add an intrinsic reward to the objective of conventional greedy policy: "utilized the idea of the uncertainty captured from the variance of value distribution" is correct. DLTV uses the distribution/quantiles/variance to estimate the upper confidence bound and use it for action selection. It's not "adding an intrinsic reward".
>
> Thank you for the clarification. Modification has been made accordingly.
>
> > our work is the first to connect stochastic policy as a solution to the problems in value-based DRL: this isn't very clear.
>
> We agree that the sentence is not clear and does not deliver any critical message. Thus, it has been removed from the text.
>
> > We believe that the findings from this paper can easily generalize to other actor-critic frameworks as well: are you sure the other alg's all have the same kind of the (three) issues?
>
> Through our paper, we intended to illustrate a correct, efficient method to extend a scalar value actor-critic framework (e.g. PPO) to the distributional perspective. The three issues are not applicable to scalar value algorithms since they do not carry a distributional representation of the value function. Nevertheless, we believe that the multi-step distributional Bellman target generation procedure using SR(\lambda) could be easily generalized to other distributional actor-critic frameworks as well since those algorithms also exploit multi-step Bellman targets such as the Monte Carlo samples, truncated trajectories, and \lambda-returns. It seems that this point was not delivered clearly through the sentence quoted in the question. Therefore we have re-written the last sentence of the Related Works section to better depict our intentions.

---

> > ### Author Response · Authors · 2020-11-19
> > **Comments for AnonReviewer1 - continued**
> >
> > > Below eq 15: do you mean Bellman optimality operator is a contraction mapping?...is not a contraction in any norm (pls confirm this is correct).
> >
> > Regarding eq.15, you are absolutely correct that the Bellman optimality operator is not a contraction mapping in any norm. We were referring to the expected Bellman operator which was not clear enough with the missing pi superscript. We have made the fix in the main text.
> >
> > > Representing Multimodality section: Here the experiment lacks a study of asymmetric models…Q value function distribution is not symmetric.
> >
> > Please refer to our answers to the question regarding the Gaussian distribution and asymmetric property.
> >
> > > Discrete and Continuous Action Spaces section: Do you use the same algorithm for both cases? Why PPO, IQAC IQAC-E GMAC are selected? How these alg's compare to distRL alg's like DQN, C51, QRDQN?
> >
> > For both the discrete and continuous action space, we used the exact same method, except for the action distribution parameterization. This was because what we want to provide in this work is a more general framework that does not depend on the type of action space. PPO was selected as a baseline for scalar-based actor-critic with stochastic policy, and as mentioned at the beginning of Section 5, the other 3 algorithms were built upon PPO. The learning curves of IQAC are shown for comparison with the recent works using Huber quantile loss, while IQAC-E and GMAC are the extensions using Dirac mixture of samples and Gaussian mixtures respectively. For your question on the relationship with value-based methods, PPO could be regarded as DQN, the scalar baseline, and IQAC is the implicit quantile version of PPO, which makes it similar to IQN. To the best of our knowledge, IQAC-E and GMAC do not have an exact equivalent.

---

> > ### Comment · AnonReviewer1 · 2020-11-22
> > **thanks for clarification**
> >
> > thanks for your feedbacks.
> >
> > Now I understand your algorithm has only one version for both discrete and continuous action space, because you used stochastic policy, this is cool.
> >
> > I also noted one of your main contribution is the return distribution you added for the multi-return signal. Reviewer 4 has mentioned other papers like Rainbow has explored it before. It helps to have a close look and discussion on this.
> >
> > In all, I'm satisfied with your feedbacks.

---

> > > ### Author Response · Authors · 2020-11-24
> > > **Related works for the multi-return signal**
> > >
> > > Thank you for pointing out the relationship with the literature on multi-return signals. During the revision period, we have included the mentioned papers and added more description on the topic in Section 2.
> > >
> > > Again, thank you for taking the time for a thorough review.

---

### Official Review · AnonReviewer4 · 2020-10-29
**Concerns with clarity, attribution and correctness**

**Rating:** 5
**Confidence:** 4

**Review:**

This paper proposes to learn a Gaussian Mixture Model of the distribution of returns and use it as the critic in an actor-critic RL agent. From what I can tell the principal novel contribution of this work is the Sample-Replacement method, in particular the observation that when paired with a GMM the replacement can be done at the level of modes of the mixture instead of individual samples. Another potential contribution is showing that the GMM can be optimized using the Cramer metric, although obviously this metric has been fairly widely studied previously.

However, this work has several problems that make it unpublishable at the moment. I'll begin with the least severe (lack of clarity and poor attribution) and then move to the much more problematic (misleading statements and factual errors).


Unclear:

Section 4.3, assumptions:
"1) the reward given a state-action pair R(x, a) follows a single distribution with finite variance"

What does this mean? Finite variance is clear, what do you mean "a single distribution".

"3) the policy follows a distribution which can be approximated by Dirac mixtures"

What does this mean? Approximated how well? Under what metric?

Equation 18, second line, \mu and \sigma seem like they should both be functions of (x, a).


Poor attribution:

Existing work has used Gaussian Mixture Models for distributional RL, as well as for the actor-critic setting (D4PG, among others).

Existing work has considered multi-step returns in distributional RL (Rainbow, Reactor, as well as almost all methods that use AC with Dist. RL). However, the Sample-Replacement method is an interesting contribution that is novel compared with this existing work.

"The distributional Bellman operator is a [...] contraction mapping in the Cramer metric space, whose proof can be found in Appendix C."

This has previously been proven in the Rowland et al. (2019) paper the authors cite, but do not attribute such a result to.


Misleading statements:

"Third, the Wasserstein distance that is commonly used in DRL does not guarantee unbiasedness in sample gradients"

While this is true for direct minimization, the quantile regression work cited in this paper does guarantee unbiased sample gradients.

"The instability issue is not present under the stochastic policy... Combining these solutions, we arrive at a distributional actor-critic..." (Much later) "One way to overcome this issue is learning value distributions under the stochastic policy and finding an optimal policy under principles of conservative policy iteration..."

The instability issue the authors reference here is that the distribution of returns, though not its mean, can be an expansion under any probability metric when applying the optimality operator. While this is an interesting topic, the authors do not actually address it or contribute towards its understanding or solution in any way. The evaluation operator was already known to be a contraction in Wasserstein (as well as for Cramer), which is the relevant operator when considering an actor-critic framework. Unlike the authors' claim that this is due to using a stochastic policy, it is in fact due to performing evaluation as opposed to optimality operators.

"Barth-Maron et al (2018) expanded DDPG by training a distributional critic through quantile regression."

This is completely incorrect, as they considered categorical distributions and Gaussian mixtures, but not quantile regression.


"The actor-critic method is a specific case of temporal-difference (TD) learning method in w hich the value function, the critic, is learned through the TD error defined by the difference..."

Actor-critic uses TD to learn the critic, but it is not a specific case of TD.

"However, the Wasserstein distance minimized in the implicit quantile network cannot guarantee the unbiasedness of the sample gradient, meaning it may not be suitable for empirical distributions like equation 13."

This is 100% false and shows a lack of understanding of multiple papers being cited in this work.

Figure 2 and "Wasserstein distance (labeled as IQ) converges to a local minimum which does not correctly capture the locations of the modes"

This seemed off to me so I went ahead and reimplemented this experiment myself. This has nothing to do with the Wasserstein distance and is exceedingly misleading to the reader. Suggestion to read the Rowland et al. (2019) paper that the authors cite for better understanding. Huber-quantiles are not quantiles. The authors learn Huber-quantiles and then treat them as quantiles and observe they look wrong. If you run IQ with the Huber parameter at 0 (corresponding to quantiles) then you get the correct (unbiased) distribution. If you instead learn Huber-quantiles and use the imputation in the Rowland you again get the right distribution.

The experimental results in the main text look promising for GMAC, but looking at the full set of results in the appendix paints a much more mixed picture.

---

> ### Author Response · Authors · 2020-11-19
> **Comments for AnonReviewer4**
>
> We thank you for your thorough review and some critical corrections on major arguments. Adhering to your advice, we made a correction to one of our arguments which claims the performance of Huber quantile is due to the property of Wasserstein distance, which is absolutely unrelated. As suggested, we also believe that the performance issue, especially the ones presented via the two-state MDP problem, is mostly due to the data conflation caused by the “type error”. To further test this, we extended the toy example to 5 states, with the early states having zero rewards. Similar to that of Rowland (2019), we were able to see the underestimation of variance in the case of the naive-Huber quantile. The performance was indeed enhanced through imputation, but not as significant as just using the quantile regression (kappa=0).
>
> Based on these observations, which we have replaced the old toy example with, we were able to change several critical issues that you have given us. First, the Wasserstein distance should only remain as our motivation, and all experimental results adhere to the characteristics of Huber quantile only. To this end, we have made clear distinctions between the Wasserstein distance and the Huber quantile loss throughout the paper, which was very misleading and wrong.
>
> To summarize, the inaccurate representation of the previous methods (such as Huber quantile) is due to the conflation as shown in Rowland et al. (2019). On the other hand, by directly parameterizing the value distribution using random samples of the distribution via implicit quantile network and minimizing the energy distance (IQE), or by predicting the parameters of the distribution via GMM and minimizing the energy distance, we were able to bypass the conflation problem and thus a more accurate representation of the distribution is learned.
>
> > Unclear
> > Section 4.3, assumptions: "1) the reward given a state-action pair R(x, a) follows a single distribution with finite variance"
>
> This statement does not add any significance to our arguments and therefore was removed.
>
> >"3) the policy follows a distribution which can be approximated by Dirac mixtures"
>
> We have changed these informal claims to an assumption on the probability measures for the value distribution itself above eq (13), so that we can appeal to the existence of dense sets in the set of finite signed Borel measures (Bogachev, 2007). The convergence can now be defined on the weak* topology, where the underlying topological vector space is defined as a 1-dim Euclidean space (the real line).
>
> > Equation 18, second line, \mu and \sigma seem like they should both be functions of (x, a).
>
> In the process of re-writing parts of section 4, this equation was no longer necessary and thus was removed.
>
> > Poor attributions
>
> All of the missing attributions have been added appropriately and other vague, misleading, and false expressions have been corrected as you have advised. Such corrections in our revised manuscript include citations and give full credits to the results in Rowland et al. (2019) and other literature mentioned in the comment. The description to Barth-Maron et al. (2018) was incorrect and was fixed in the main text.
>
> Regarding the experimental results, we have included the full learning curve of 61 atari games and the human-normalized scores. Results from IQAC-E have also been added to Fig. 9 and Table 5 in the Appendix. Hopefully, these would provide a much clearer picture of the performances in the case of discrete action space. Results from additional seeds along with results from IQAC will be added as soon as they are ready.
>
> **References**
>
> [1] Mark Rowland, Robert Dadashi, Saurabh Kumar, R ́emi Munos, Marc G. Bellemare, and Will Dab-ney. Statistics and samples in distributional reinforcement learning. ICML, 2019.
>
> [2] Vladimir I Bogachev. Measure theory, volume 1. Springer Science & Business Media, 2007.

---

> > ### Comment · AnonReviewer4 · 2020-11-24
> > **Thank you for the revisions**
> >
> > Reading through the latest revision, it is clear the authors have made a real attempt to address many of the concerns I have raised.
> >
> > The new results in Figure 2 seems to be making a related point to that of Rowland et al. (2019), and the mode collapse seen here does seem like a reasonable consequence. I am a bit confused by the poor estimates produced by your IQ-imputation example. If time permits, could you include some discussion of the observed effect? Can you also verify that when setting kappa = 0 you observe a reasonable approximation of the true distribution?
> >
> > I will be increasing my rating, but want to do a more thorough pass through the updated paper and discuss with other reviewers before making the update.

---

> > > ### Author Response · Authors · 2020-11-24
> > > **On the estimate from IQ-imputation**
> > >
> > > >  I am a bit confused by the poor estimates ... Can you also verify that when setting kappa = 0 you observe a reasonable approximation of the true distribution?
> > >
> > > We believe a possible cause for the poor estimate is the small scale of rewards: the true return distributions in Fig. 2 of Rowland et al. (2019) and Fig. 5(d) of our paper both have a larger scale, and you can observe that in Fig. 5(d) the Huber quantile regression with imputation can approximate the true distribution as well. Furthermore, the results in Fig. 5(b) and (e) show that the loss can converge to a reasonable amount when using the quantile regression (kappa = 0), with and without imputation.
> > >
> > > Again, thank you for taking your time and keen advice during the review.

---

### Official Review · AnonReviewer3 · 2020-10-30
**Interesting novel ideas but some concerns w.r.t evaluation.**

**Rating:** 6
**Confidence:** 2

**Review:**

This paper proposes a distributional Actor critic framework (GMAC) based on GMM, Actor critic and Cramer distance.
Authors introduce SR(λ) a distributional version of the λ-return algorithm and to minimize the Cramer distance - as opposed to minimizing the Wasserstein distance using Huber quantile regression- between the value distribution, this helps in obtaining unbiased sampled gradients, this is shown to be more effective in preserving modality that can provide extra information in sparse-reward exploration tasks as well as more stability during training. Finally, authors choose to parametrize value distributions as a GMM this provides a closed-form to the energy distance (eq19) which can reduce computational costs achieving very close numbers to PPO.

Clarity: The paper is easy to follow and well written. The motivations are quite clear from the beginning. The paper would have benefited from highlighting why GMAC specifically is suitable to handle discrete and continuous actions and the challenges behind each case (see q1).

Novelty: This work introduces many novel aspects, importantly the use of the Cramer + GMM for getting an unbiased sample gradient plus computational efficiency.

Experiments and significance of the empirical results:
Authors evaluate GMAC using a two-state MDP, a set of discrete and continuous action space tasks. the majority of the results presented show convincing improvements of GMAC over IQAC and the PPO baselines. IQN + energy distance (IQAC-E) shows improvements in capturing the modality over IQAC confirming the intuition behind the proposed use of Cramer distance. Both GMAC and IQAC-E show improvements in the computational costs. However, I do have some concerns considering the selection of the displayed examples and table of results in the appendix including only the baseline PPO (Table 5) (see q2&q3).


Questions:
q1: in Figure 7, the PyBullet learning curves, In 3/5 of the learning curves IQN + Huber quantile (IQAC) seems to be performing on par or better than GMAC. this makes me wonder what conceptually makes GMAC specifically suitable for both Discrete and continuous action spaces.

q2: Since there are no space limitations in the appendix. In Figure 6, I am wondering the reasons behind displaying only the learning curves of 8 selected games?

q3: Could you In Table 5 display the Average scores of IQAC and IQAC-E. Some of the reported results of GMAC are inferior to those (QR-DQN & IQN) which are reported in (Dabney et al 2018a) If those results comparable, It would be better to put them side by side for comparison to confirm the claims wrt performance superiority.

---

> ### Author Response · Authors · 2020-11-19
> **Comments for AnonReviewer3**
>
> Thank you for the clear, constructive review.
>
> While we aim to answer all of the questions that you have raised, we would like to start off with a comment regarding the Wasserstein distance and the Huber quantile loss. As mentioned in the reviews, solving an optimization problem using the Huber quantile function is not the same as using the Wasserstein distance. It was wrong, and misleading to treat minimizing the Huber quantile loss as if it is equal to minimizing the Wasserstein distance. Therefore, we have corrected our statement about the cause of biased representations, as the conflation between statistics and samples when using the Huber quantile loss.
>
> > Question1: in Figure 7, the PyBullet learning curves...both Discrete and continuous action spaces.
>
> It seems that our expression of handling both discrete and continuous cases was unclear. The main goal of the paper is that we wish to propose an efficient value distribution learning algorithm that correctly captures the distribution so that the distribution can be utilized in a much more broad manner in the future. As an example, we ran an experiment that uses the extra information that can only be gained from the distributional form of the value function to solve a known hard-exploration task of “Montezuma’s Revenge” in atari. Furthermore, the experiments with the toy task and modality were also to show that the distribution is correctly captured. On top of this, we have observed that the human-normalized mean score of 57 atari games has also increased, just by expanding the scalar values to distributions. On the other hand, another goal was to create an algorithm that can be applied in both the discrete action space and the continuous action space. To this end, we applied the exact same algorithm to the continuous control tasks. The score did not show any significant increase, but it ensures that our algorithms can work as a ground point from which modifications that utilize the distributional characteristics well in the continuous control tasks can be built upon.
>
> Therefore, it was unclear to treat the atari tasks and the continuous tasks on the same ground and thus we have changed our wording in the main text to emphasize that our algorithm can be applied to both action spaces without much modification.
>
>
> > Question2: Since there are no space limitations in the appendix. In Figure 6, I am wondering the reasons behind displaying only the learning curves of 8 selected games?
>
> There have been several concerns regarding this so we have included the full learning curves and human-normalized scores for 61 atari games for PPO, IQACE, and GMAC. Through the original choice of the 8 games, we attempted to depict a wide variety of game types, from the ones which have shown score improvements in previous distribution RL literature (Breakout, Beamrider, Gravitar) to ones that may contain stochasticities in its nature, like Gopher with the movement of NPC being stochastic at times, to hard tasks which require explorations, like Montezuma’s Revenge and Venture. However, this was not clear enough just by the tasks themselves so we included the learning curves of 61 atari games and the human-normalized scores. Further results from extra seeds in these experiments are scheduled to be added as soon as they are complete.
>
>
> > Question3: Could you In Table 5 display the Average scores...to confirm the claims wrt performance superiority
>
> We have added the scores for IQAC-E in table 5. For IQAC, we did not consider running the full test since the imputation problem is apparent through past literature (Rowland 19) and our toy example of 5-MDP. This becomes a more serious problem when n-step approximations like TD(\lambda) are used, especially when n is large, due to the conflation from successive Bellman operations. However, we acknowledge that the results from IQAC would be helpful for comparison, thus are also to be added as soon as they are ready.
>
>
> **References**
>
> [1] Mark Rowland, Robert Dadashi, Saurabh Kumar, R ́emi Munos, Marc G. Bellemare, and Will Dabney. Statistics and samples in distributional reinforcement learning. ICML, 2019.

---

### Comment · Area_Chair1 · 2020-11-22
**Discussion time**

Dear Reviewers,

The authors have provided a detailed response and uploaded their revised manuscript. Would you please take a careful look at their response and revision? Please respond to the authors and update your review accordingly.

Thanks,
AC

---

### Comment · Area_Chair1 · 2020-11-23
**problematic statements and notations, questionable comparison**

Given the disagreement between the reviewers, I have read the paper carefully. While I know the related literature well, I have found the paper difficult to understand. There are a number of statements that appear problematic to me and the notations are confusing in quite a few places. The lack of a detailed algorithm box is also not helping. Moreover, It is unclear why existing baselines related to distributional RL have not been included for comparison. Below I list some examples:

1. "While Wasserstein distance is biased, in practice, directly minimizing the Wasserstein distance is often infeasible and thus some of the exemplary works (Dabney et al., 2018b;a) of deep DRL minimizes the Huber quantile regression loss (Huber, 1964) instead"

What do you mean "Wasserstein distance is biased?" I suspect what you are trying to say is that the Wasserstein distance estimated with empirical distributions is biased. Please clarify.

"directly minimizing the Wasserstein distance is often infeasible" Again this statement is confusing. Note Wasserstein distance in 1D, which is the case you are dealing with in distributional RL, is actually quite simple to estimate. Suppose you have two empirical distributions, both supported on N iid samples, then their Wasserstein1 distance can be simply computed by sorting them and then aggregating element-wise absolute difference.


2. "However, as proven in Rowland et al. (2019), representing a distribution using the Huber quantiles instead of samples can
lead to conflation problems"

Could you please elaborate the definition of "conflation problems"?

3. "However, previous works concentrated on extending a specific actor-critic framework to
the distributional setting. Therefore, we aim to suggest methods which may be easily adopted in the
process of expanding a scalar value methods to a distributional perspective, along with an attempt to
address the previously mentioned issues present in the value-based ditributional algorithms."

To my understanding, the previous work in distributional RL is primarily focused on modeling the distribution of the discounted cumulative return of action $a$ under state $s$, whose expectation is the action-value function $Q(s,a)$. Could you please elaborate on why taking a distributional view of the value function is preferred to taking a distributional view of the action-value function?  Do you have empirical results to verify the advantages?

4. Eq 13, $\frac{1}{m}\sum_{i=1}^m \delta_{z_t^{(n)}}$

Shall $\delta_{z_t^{(n)}}$ be $\delta_{z_t^{(i)}}$? Similar confusion notation in Eq. 20.

5. Related to comment 3, could you further clarify why it is justifiable to not include existing distributional RL-based baselines, such as QR-DQN and IQN, for comparison?

---

> ### Author Response · Authors · 2020-11-23
> **Comments for AC1**
>
> Thank you for taking the time to review our paper. We hope that this comment will clarify your questions.
>
> > The lack of a detailed algorithm box is also not helping.
>
> Thank you for pointing out. We acknowledge that the pseudocode for Sample-Replacement (SR($\lambda$)) algorithm does not show an overall picture for the learning process, thus we added a more detailed description of GMAC in Appendix E.
>
> > 1. What do you mean "Wasserstein distance is biased?" I suspect what you are trying to say is that the Wasserstein distance estimated with empirical distributions is biased. Please clarify.
>
> We meant to say “Wasserstein has biased sample gradients”. The original expression was unclear and a correction has been made in the main text as you have suggested.
>
> > "directly minimizing the Wasserstein distance is often infeasible" Again this statement is confusing. Note Wasserstein distance in 1D, which is the case you are dealing with in distributional RL, is actually quite simple to estimate. Suppose you have two empirical distributions, both supported on N iid samples, then their Wasserstein1 distance can be simply computed by sorting them and then aggregating element-wise absolute difference.
>
> You are absolutely correct that when both the source and the target empirical distributions have the same N samples, the empirical Wasserstein distance can be easily derived. However, as pointed out in the answer to the first question, in general practice of training neural networks, the use of Wasserstein distance is not preferred due to its incompatibility with stochastic gradient descent from the biased sample gradient. Our expression was not clear about this point so we have corrected it in the main text along with the answer to the first question.
>
> > 2. Could you please elaborate the definition of "conflation problems"?
>
> The conflation problem comes from the type mismatch between samples and statistics as pointed out in Rowland et al. (2019). When applying the Bellman operator, it is crucial to make sure that the operation is taken on “samples” and not “statistics”. In other words, when the network outputs a set of statistics, e.g. quantiles trained using Huber quantile loss, an appropriate imputation strategy (Rowland et al., 2019) must be applied to reformulate the empirical samples before applying Bellman operations. By using the Energy distance-based loss as proposed in our paper, this problem is avoided by directly predicting the samples of the distribution, which effect can be seen through presented empirical results.
>
> > 3. To my understanding, the previous work in distributional RL is primarily focused on modeling the distribution of the discounted cumulative return of action $a$ under state $s$...Do you have empirical results to verify the advantages?
>
> The illustration of our algorithm in the main text focuses on the value function instead of the action-value function, due to the choice of our baseline (PPO), not because we are claiming that using the state value is theoretically superior to using the state-action value. So we did not provide an empirical result to verify the advantages of one over the other.
> We emphasize that some of the problems, such as the instability issue, are present in the value-based methods, e.g. C51, QR-DQN, and IQN, from using the Bellman optimality operation. In this case, the restriction to action type (discrete vs. continuous) also applies. Because value-based methods, in general, contain such problems, we would like to solve them by using a policy-based algorithm of the actor-critic method. Within the scope of actor-critic methods, deterministic policy gradient methods like the D4PG (Barth-Maron et al., 2019) have the action type limitation, in practice, as pointed out. As a result, we decided to use a stochastic policy gradient algorithm proposed in PPO (Schulman et al. 2017) as our baseline. PPO bases its policy update on the advantage, which only requires learning of the state value function, instead of the state-action value, which allows it to be independent of the action type.
> However, the parameterizations and the losses provided in our paper may be applied to other methods, such as the value-based method of IQN. It would be interesting to see the result, but this is outside the scope of solving the problems that we have mentioned, and thus we leave it as future work.
>
> > Eq. 13 and 20
>
> $n$ is a random variable that has Geo($1-\lambda$) as its probability distribution as stated below each equation. The use of alphabet n and its lower case was a bit confusing but we kept it as it is to be consistent with the algorithmic description of the TD($\lambda$), from which we derive SR($\lambda$). We hope that this explanation clears up the unclarity bit more.

---

> > ### Author Response · Authors · 2020-11-23
> > **Comments for AC1 - continued**
> >
> > > Related to comment 3, could you further clarify why it is justifiable to not include existing distributional RL-based baselines, such as QR-DQN and IQN, for comparison?
> >
> > It is due to the differences between the value-based methods and our methods, which we will explain below. Therefore we decided to not directly compare our results to that of IQN, QR-DQN, etc., rather we focused on presenting the differences among our methods in a clear way. We also emphasize that the goal of our framework is to capture the right modality (shape) of value distributions, not to be better than SOTA.
> >
> > Continuing from the answer to comment 3, we first acknowledge that when directly comparing the resulting scores of each atari game, IQN has higher scores in more games compared to GMAC while the mean human-normalized score of GMAC is higher. This may be due to the difference between the foundational algorithms of each method, i.e. between the value-based method with experience replay and a stochastic policy gradient method with parallel environments.
> >
> > This difference further diverges in the evaluation process of the performance where best human-normalized score is used for each game (Table 1 of Dabney et al. 2018) in IQN, while for stochastic policy gradient method like PPO, often the average score of 100 most recent episode is reported due to its stochastic nature.
> >
> > This behavior is clearer when directly comparing the results from IQN (Figure 4), where the full learning curve of the mean human-normalized score is reported to max around {~800, >600, <600}% for Rainbow, IQN, and QR-DQN respectively. On the other hand, the end of the mean human-normalized score of 100 recent episodes for PPO, IQAC-E, and GMAC are about {980, 920, 1200}%, which shows a significantly different trend.
> >
> > This difference in behaviors comes from the fact that the atari games for which each category of algorithms has extreme scores differ. In both cases, the total number of frames used to train was the same. To mention the median human normalized-scores, it was ~{<200, 150, 130} for Rainbow, IQN, and QR-DQN, while for PPO, IQAC-E, GMAC it was ~{125, 148, 136}, respectively. Meanwhile, we include the full learning curve of human-normalized scores in Figure 3 for curious readers.
> >
> >
> > **References**
> >
> > [1] Mark Rowland, Robert Dadashi, Saurabh Kumar, R ́emi Munos, Marc G. Bellemare, and Will Dabney. Statistics and samples in distributional reinforcement learning. ICML, 2019.
> >
> > [2] Gabriel Barth-Maron, Matthew W. Hoffman, David Budden, Will Dabney, Dan Horgan, Dhruva TB, Alistair Muldal, Nicolas Heess, and Timothy Lillicrap. Distributed distributional deterministic policy gradients. ICLR, 2018.
> >
> > [3] John Schulman, Filip Wolski, Prafulla Dhariwal, Alec Radford, and Oleg Klimov. Proximal policy optimization algorithms. arXiv preprint arXiv:1707.06347, 2017.
> >
> > [4] Will Dabney, Georg Ostrovski, David Silver, and Remi Munos. Implicit quantile networks for distributional reinforcement learning. ICML, 2018a.

---

### Decision · Program_Chairs · 2021-01-07
**Final Decision**

**Decision:**

Reject

**Comment:**

The paper proposes a distributional perspective on the value function and uses it to modify PPO for both discrete and continuous control reinforcement learning tasks. The referees had noticed a number of wrong/misleading statements in the initial version of the submission, and the AC had also pointed out several problematic statements in a revised version. While the authors had acknowledged these mistakes and made appropriate corrections, there are several places that still need clear improvement before the paper is ready for publication. The paper seems to introduce a novel actor-critic algorithm. However, the correctness of its key step, the  SR($\lambda)$ algorithm, has not been rigorously justified. For example, it is unclear how the geometric random variables would arise in that algorithm. For experiments, the AC seconds the comments provided by Reviewer 2 during the discussion: "The empirical comparisons are overall still lacking: for the smaller-scale experiments, whilst the authors have been actively engaged in improving these comparisons during the rebuttal, at present, they are still in need of updating to make a fair comparison, for example in terms of the number of parameters included. The authors have acknowledged this, although the rebuttal period ran out before they were able to post new plots. The large-scale empirical results are still lacking reasonable baselines against existing distributional RL agents."